# Context-dependent enhancer function revealed by targeted inter-TAD relocation

Christopher Chase Bolt[1,2 ✉], Lucille Lopez-Delisle [1], Aurélie Hintermann[2], Bénédicte Mascrez [2], Antonella Rauseo[3,4], Guillaume Andrey [3,4] & Denis Duboule [1,2,5 ✉]

The expression of some genes depends on large, adjacent regions of the genome that contain multiple enhancers. These regulatory landscapes frequently align with Topologically Associating Domains (TADs), where they integrate the function of multiple similar enhancers to produce a global, TAD-specific regulation. We asked if an individual enhancer could overcome the influence of one of these landscapes, to drive gene transcription. To test this, we transferred an enhancer from its native location, into a nearby TAD with a related yet different functional specificity. We used the biphasic regulation of *Hoxd* genes during limb development as a paradigm. These genes are first activated in proximal limb cells by enhancers located in one TAD, which is then silenced when the neighboring TAD activates its enhancers in distal limb cells. We transferred a distal limb enhancer into the proximal limb TAD and found that its new context suppresses its normal distal specificity, even though it is bound by HOX13 transcription factors, which are responsible for the distal activity. This activity can be rescued only when a large portion of the surrounding environment is removed. These results indicate that, at least in some cases, the functioning of enhancer elements is subordinated to the host chromatin context, which can exert a dominant control over its activity.

[1] School of Life Sciences, Ecole Polytechnique Fédérale de Lausanne (EPFL), 1015 Lausanne, Switzerland. [2] Department of Genetics and Evolution, Faculty of Sciences, University of Geneva, 30 quai Ernest Ansermet, 1211 Geneva, Switzerland. [3] Department of Medical Genetics, Faculty of Medicine, University of Geneva, Rue Michel Servet 1, 1211 Geneva, Switzerland. [4] Institute of Genetics and Genomics in Geneva (iGE3), University of Geneva, Geneva, Switzerland. [5] Collège de France, 11 Place Marcelin Berthelot, 75231 Paris, France. ✉email: ccbolt@gmail.com; denis.duboule@epfl.ch

Genes with important functions during vertebrate development are frequently multifunctional, as illustrated by their pleiotropic loss-of-function phenotypes. This multi-functionality generally comes from the accumulation of tissue-specific enhancer sequences[1] around the gene. At highly pleiotropic loci, the collection of enhancers can be distributed across large genomic intervals, which are referred to as regulatory landscapes[2,3]. Such landscapes often correspond in linear size to topologically associating domains (TADs)[4–6], defined as chromatin domains wherein the probability of DNA-to-DNA interactions is higher than with neighboring domains. These collections of enhancers likely reflect an evolutionary process whereby the emergence (or modification) of morphologies was accompanied by novel regulatory sequences with the ability to be bound by tissue-specific factors and hence alter the transcription of nearby target genes[7].

This model, however, does not explicitly account for the potential influence of local genomic context to modulate interactions between promoters and nascent enhancers, thus adding another layer of complexity to our understanding of developmental gene regulation. For instance, many developmental gene loci are initially decorated with the chromatin mark H3K27me3 deposited by the Polycomb complex PRC2, before their activation[8], possibly as a way to maintain silencing or limit transcription to very low levels. These chromatin modifications disappear along with gene activation and are re-deposited once the gene and its enhancers are progressively decommissioned, thus preventing inappropriate transcription that may cause severe developmental problems (see ref. [9]). This indicates that enhancer or promoter function can be affected by their local chromatin environments.

In the case of large regulatory landscapes containing many enhancers with similar tissue specificity, the question arises as to how their activation and decommissioning are coordinated and enforced over very large genomic intervals to overcome any contradictory inputs. Two scenarios can be considered in this context. The first possibility is that each enhancer sequence within the landscape independently responds to information delivered solely through the factors bound to it in a sequence-specific manner. Such a mechanism would act entirely in *trans* and its coordination would be ensured by the binding of multiple similar positive or negative factors. In the second scenario, a higher level of regulation is imposed by the general environment of the landscape, for example corresponding to a TAD-wide positive or negative regulation, which would tend to dominate individual enhancer-specific controls in order to minimize dangerous misexpression events.

To address this issue, we used the development of the vertebrate limb as a paradigm and, in particular, the essential function of *Hox* genes in patterning and producing the main pieces of tetrapod appendages[10]. The emergence of the tetrapod limb structure was a major evolutionary change that facilitated animal migration onto land. A key step in this process was the acquisition of a fully developed distal part (hands and feet), articulating with the more ancestral proximal parts of the limbs (e.g., the arm and the forearm). During development, the distal piece requires the specific function of several key genes, amongst which are *Hoxa13* and *Hoxd13*. While the absence of either gene function induces a moderate phenotype in hands and feet[11,12], the double loss-of-function condition leads to the agenesis of these structures[12] thus suggesting a critical function for these two genes both during the development of distal limbs and its evolutionary emergence (see refs. [13,14]). In this context, the regulatory mechanisms at work to control the expression of both genes in most distal limb cells were carefully analyzed either for *Hoxa13*[15,16], or for *Hoxd13*[3,17], and found to be quite distinct from one another.

In the case of *Hoxd* genes, the evolutionary co-option of *Hoxd13* into both the distal limbs and the external genitals was accompanied by the emergence of a novel TAD (C-DOM) containing multiple enhancers specific to either tissue or common to both[17–20]. This TAD is inactive until late in development when limb cells with a distal fate start to appear. In contrast, the expression of other *Hoxd* genes (*Hoxd9*, *Hoxd10*, and *Hoxd11*), which are all critical for the formation of more proximal parts of the limb[21], are controlled by enhancers acting earlier and located in the adjacent TAD (T-DOM)[22]. The *HoxD* cluster lies in between these two TADs and contains several CTCF binding sites that create an interaction boundary between the two adjacent chromatin domains, insulating *Hoxd13* and *Hoxd12* from the rest of the gene cluster[23]. In the early stages of limb bud formation, *Hoxd9*, *Hoxd10*, and *Hoxd11* are activated by the regulatory landscape located telomeric to the cluster (T-DOM) leading to the patterning and growth of long bones of the arms and legs. Subsequently, in a small portion of cells at the posterior and distal end of the early limb bud, enhancers located in the centromeric regulatory landscape (C-DOM) are switched on, driving the expression of 5′-located *Hoxd* genes in the nascent hands and feet[22].

The operations of these two TADs are mutually exclusive and as soon as C-DOM starts to upregulate the expression of *Hoxd13* in distal cells, enhancers within T-DOM are decommissioned and a large part of this chromatin domain becomes decorated by H3K27me3 marks[22]. This switch in TAD implementation involves the HOX13 proteins themselves since their production in response to C-DOM enhancers participates in the repression of T-DOM enhancers, likely through direct binding[24,25]. In parallel, HOX13 proteins positively regulate C-DOM enhancers, thus re-enforcing the functional switch between the two TADs. In the absence of both HOXA13 and HOXD13, C-DOM is never activated whereas T-DOM enhancers continue to function due to the absence of both decommissioning and H3K27me3 coverage[24]. Therefore, in this context, HOXD13 shows the properties of both a transcriptional activator and a transcriptional repressor at different times and in different chromatin environments.

At the time when *Hoxd13* becomes activated by distal limb (digit) enhancers, it is critical that all proximal limb (forearm) enhancers are rapidly switched off, to prevent the latter may act on the former, a situation that was shown to be detrimental to limb morphology[26]. This bimodal regulatory situation thus provides a paradigm to assay for the presence of a 'context-dependent' repression of enhancer activity. Accordingly, we set out to introduce into the proximal limb-specific T-DOM domain, the strongest distal enhancer normally working within C-DOM to activate *Hoxd13* in digit cells, asking whether this distal limb regulatory sequence would still be able to exert its function when relocated into a TAD where limb enhancers are being repressed in distal cells. We report that this enhancer element, which is functionally very penetrant when introduced at various random positions by non-targeted transgenesis, loses most of its distal limb activity when recombined within T-DOM, even though it continues to recruit the HOX13 factors, which are essential for its function in distal limb cells. We further show that part of the distal limb activity is restored when large portions of T-DOM are deleted in *cis*. We conclude that the function of this enhancer is inhibited by an *in cis* mechanism acting at the level of an entire chromatin domain, suggesting the existence, in this particular case, of a level of regulation higher than that of the enhancer sequences themselves.

## Results

**A distal limb-specific enhancer.** To evaluate how a tissue-specific enhancer would behave when relocated into a different chromatin

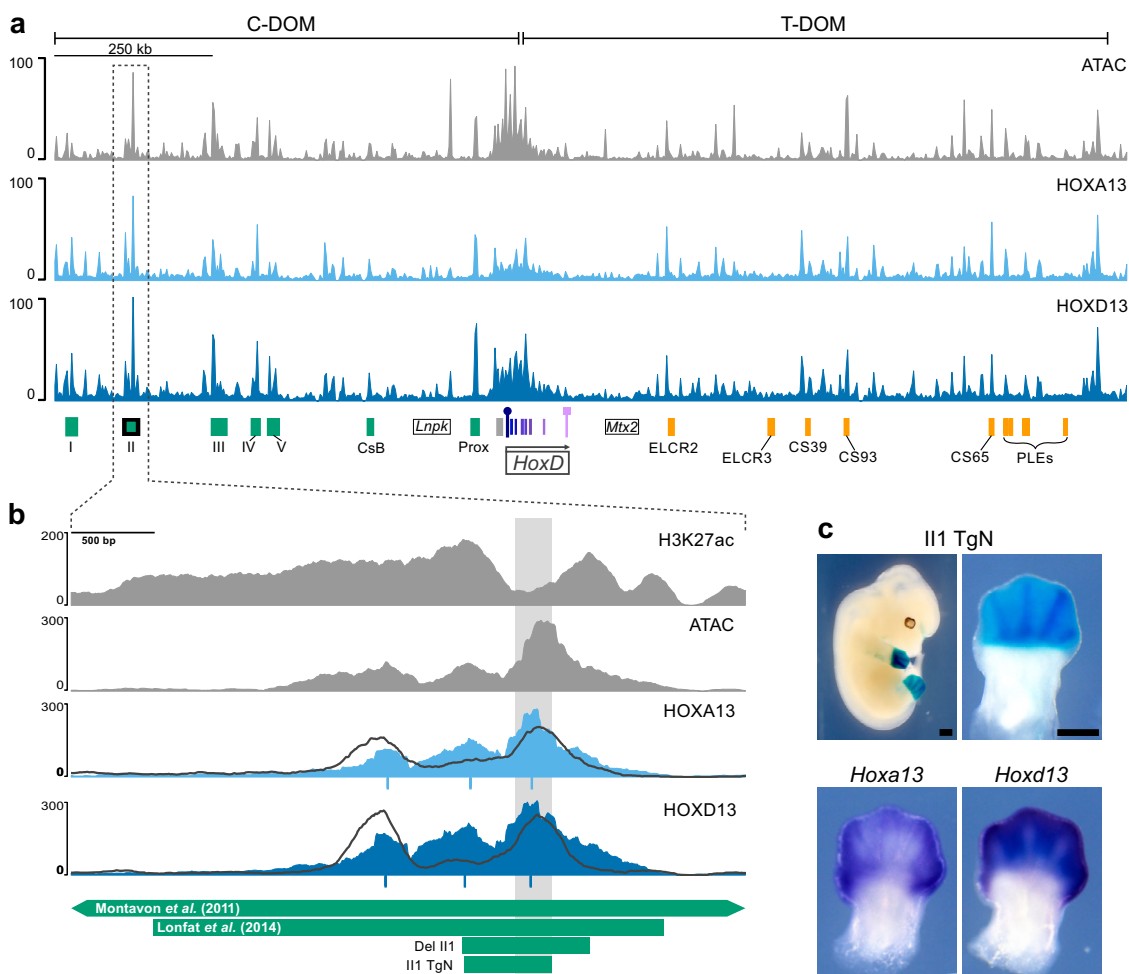

**Fig. 1 Identification of a distal limb bud-specific enhancer. a** ATAC-seq profile (top) and binding profiles of the HOXA13 (middle) and HOXD13 (bottom) transcription factors by CUT&RUN, using E12.5 wild-type distal forelimb cells, covering the entire *HoxD* locus, including the two flanking TADs C-DOM and T-DOM (mm10 chr2:73950000-75655000). Green rectangles below are distal limb enhancers in C-DOM and orange rectangles are proximal limb enhancers in T-DOM. The *Hoxd* gene cluster is indicated at the center (the 5′ to 3′ direction of transcription is indicated by a black arrow) and the *Lnpk* and *Mtx2* genes are indicated as rectangles with black borders. The *Hoxd13* gene is on the centromeric end of the cluster and indicated by a purple rectangle with a circle on top, whereas *Hoxd1* is telomeric and indicated by a square top. The C-DOM Island II enhancer[17] is a green rectangle with a black border and its corresponding signal peaks are surrounded by a dashed vertical box. **b** Magnification of the same tracks as above centered around the Island II enhancer with the H3K27ac ChIP-seq signal (top) in E12.5 distal forelimbs[23]. The overlying profiles indicated by dark gray lines are from ChIP-seq using E11.5 whole limb buds[25] and are shown for comparison. Below the CUT&RUN profiles are the MACS2 peak summits for the corresponding CUT&RUN samples. The green rectangles below indicate regions described as Island II in[17] or in[27]. The Del II1 shows the region deleted in this work (Supplementary Fig. 1-2) and II1 TgN is the transgene used panel in the next panel. **c** *LacZ* staining pattern produced by the II1:*HBB:LacZ* (II1 TgN) enhancer reporter transgene at E12.5, showing high specificity for distal limb cells (top). Below are whole-mount in situ hybridizations for *Hoxa13* and *Hoxd13* in wild-type E12. 5 forelimbs for comparison. Scale bars are 0.5 mm.

and regulatory context, we searched for a candidate enhancer element that would display the strongest possible specificity for developing distal limb cells at the *HoxD* locus. We performed ATAC-seq in wild-type E12.5 distal limb cells and compared the signals with a previously reported H3K27ac ChIP-Seq dataset[23]. Because distal limb enhancers located at *Hox* loci were shown to colocalize with the binding of HOX13 proteins as assayed in ChIP-seq experiments[24,25], we used a CUT&RUN approach for both HOXA13 and HOXD13 (together referred to as HOX13) to more precisely delineate enhancer elements controlled, at least in part, by these transcription factors.

We identified a small element within a region previously described as Island II, one of the islands of the C-DOM regulatory archipelago[17,27]. This element was strongly bound by both transcription factors and accessible as judged by ATAC-Seq, suggesting that it is an active enhancer element in distal limb cells

and that it controls the expression of the most posterior *Hoxd* genes there (Fig. 1a). Upon closer inspection, we found that the peak from CUT&RUN experiments matched precisely with the peak observed in a previously reported E11.5 whole limb bud ChIP-Seq experiment for HOXA13 and HOXD13 (Fig. 1b, dark gray lines)[25]. It also matched a small DNA fragment that is highly conserved across tetrapod amniote species but thus far absent from all evaluated anamniote genomes including that of coelacanth (Supplementary Fig. 1-1a; dashed box). This element had all the hallmarks of an active distal limb enhancer controlled by HOX13 transcription factors and hence we called it enhancer element II1.

To test if this short element indeed carries the enhancer activity reported for the large versions of Island II (Fig. 1b and Supplementary Fig. 1-1a; green rectangles below), we cloned the sequence (532 bp, mm10 chr2:74075311-74075843) and

constructed a reporter transgene where the enhancer is located 5′ to the *HBB* promoter and the *LacZ* gene (Fig. 1c, II1 TgN), and injected it into mouse pronuclei. Five founder animals were identified and then crossed with wild types. Embryos were collected at E12.5 and stained for *LacZ*. Four of the founders transmitted the transgene and produced strong distal limb-specific staining (Fig. 1c, top). One of the four transmitting founders also had low staining levels in nearly the entire embryo and another founder produced staining in the central nervous system. The four animals transmitting limb staining displayed a pattern closely matching the expression domains of both *Hoxd13* and *Hoxa13* (Fig. 1c). The variation observed in staining in other tissues likely resulted from different integration sites.

We also generated a version of the transgene containing a *GFP* reporter to better evaluate the dynamic of enhancer activity during the time course of limb formation (Supplementary Fig. 1-1b). The earliest stage where GFP fluorescence was observed was at E10, which quickly became very apparent by E10.5. The fluorescence was observed in the most distal and posterior portion of the limb bud mirroring native *Hoxd13* expression (see[26]), and was never observed in the proximal limbs. This expression persisted until at least E13.5.

**Deletion of the II1 enhancer sequence.** The C-DOM regulatory landscape contains multiple enhancer elements that are necessary to produce a robust activation of *Hoxd* genes in the distal limbs and genitals[17,18] and our ATAC-Seq and CUT&RUN experiments showed that element II1 is among the most strongly bound and accessible elements throughout the C-DOM (Fig. 1a). Because of this, we anticipated that it may make a measurable contribution to the transcription of *Hoxd* genes in the distal limb. To determine what effect this element contributes to the global regulatory activity of C-DOM, we used CRISPR/Cas9 to delete the II1 enhancer element (Fig. 1b and Supplementary Fig. 1-1a; green rectangle Del II1). Several founders were obtained and we produced embryos homozygous for this ($HoxD^{DelII1}$) deletion. We collected embryos at E12.5, measured their levels of distal limb *Hoxd* mRNAs by RT-qPCR and looked at the transcript distribution by in situ hybridization. Using both approaches, we did not observe any significant change, neither in the level of *Hoxd* gene transcription in the distal limbs of embryos missing the II1 enhancer, nor in the spatial distribution (Supplementary Fig. 1-2a, b). While this result was somewhat surprising given how strong the signal for H3K27ac, ATAC, and HOX13 binding are, a similar lack of effect was observed when single genital enhancers were deleted within this same C-DOM TAD. However, removing several such enhancers had a cumulative effect thus demonstrating the regulatory resilience of this regulatory landscape[18].

**Targeted insertion of a C-DOM enhancer into T-DOM.** The T-DOM TAD contains multiple enhancers (Fig. 2a, orange rectangles), which activate *Hoxd* genes in proximal limb cells, and the deletion of T-DOM abolishes all *Hoxd* gene transcripts from the proximal pieces of the growing limb buds[22,26]. In distal limb cells, these elements are no longer at work and the entire T-DOM is switched off at the same time as enhancers within C-DOM are activated, including the II1 element. HOX13 proteins bind throughout the T-DOM and are associated with the decommissioning of the proximal limb enhancers. When HOX13 factors are not present the decommissioning does not occur and the 'proximal' enhancers continue operating in distal cells[24]. In fact, the T-DOM proximal enhancers CS39 and CS65 also operate in distal cells when introduced randomly into the genome as transgenes[24], suggesting that the T-DOM environment represses

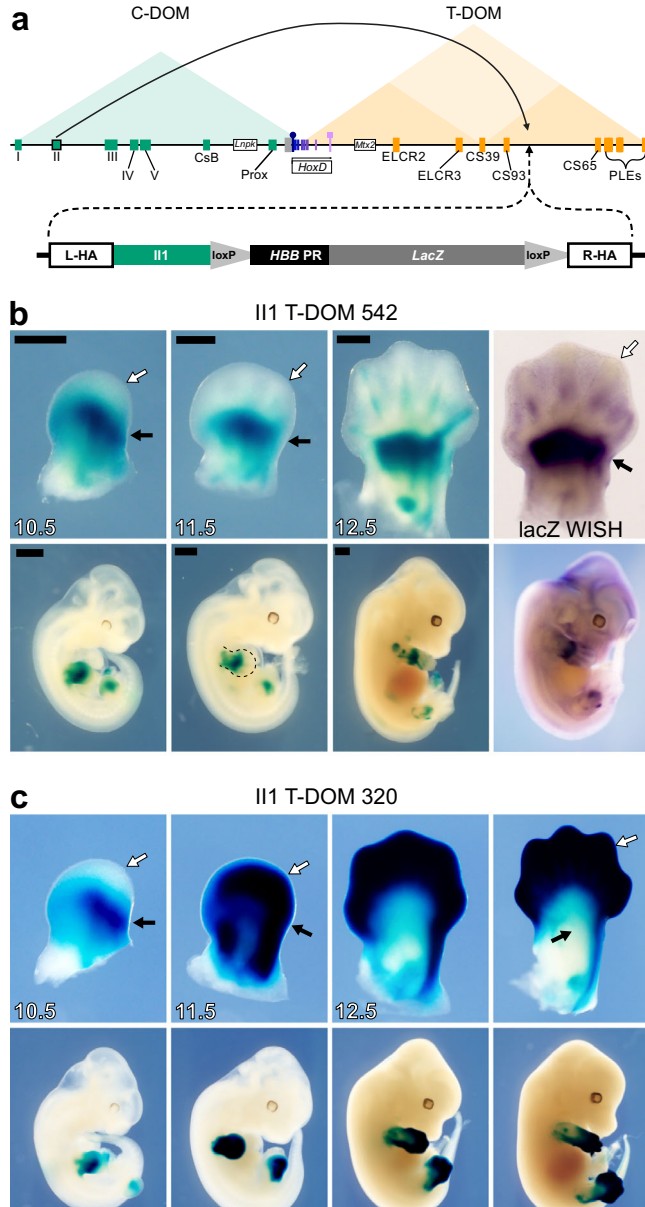

**Fig. 2 Targeted recombination of the II1 transgene into T-DOM. a** Scheme of the *HoxD* locus with the triangles indicating the extent of both the C-DOM (green) and T-DOM (orange) TADs. Enhancers are green (distal limbs) or orange (proximal limbs) rectangles and the *HoxD* cluster is boxed. The arcing arrow on top indicates the origin and new location of the island II enhancer transgene into T-DOM. Below is a map of the II1:*HHB:LacZ* construct containing left (L-HA) and right (R-HA) homology arms, the II1 enhancer element, and the *HBB* promoter with a *LacZ* reporter gene. **b** B-galactosidase staining time course and *LacZ* mRNA (right panel) of the single-copy II1 T-DOM 542 founder line. At E10.5 and 11.5, staining is strong in the proximal limb (white arrow) while absent in the distal portion (black arrow). By E12.5 weak staining appears in the digit mesenchyme of the distal limb. The WISH for *LacZ* mRNA confirms that distal limb staining comes from transcription in distal limb cells rather than from stable B-galactosidase activity. **c** B-galactosidase staining in the multi-copy II1 T-DOM 320 founder line is stronger and more distal, except at the earliest time point where it is more similar to line 542. Black scales bars are 0.5 mm.

the function of such sequences in distal cells and that this mechanism may involve the binding of HOX13 factors.

To further challenge this repressive effect and see whether it would dominate over the strong distal specificity of an enhancer that normally operates in distal cells, we set up to relocate a single copy of the enhancer II1 into T-DOM. We used the same transgenic construct (Fig. 1, II1 TgN) used to test the enhancer element by random insertion transgenesis (Fig. 1c), but we attached homology arms to the 5′ and 3′ ends to target insertion to a region of T-DOM in between—yet at a distance from—two strong proximal limb enhancers, CS39 and CS65[22,24]. This region was also selected because it had low levels of the Polycomb Group histone H3 modification H3K27me3, such that H3K27me3 short distance spreading (see ref. [28]) would presumably not directly impact the inserted element. Fertilized eggs were injected with both the targeting construct and various CRISPR components (Fig. 2a and Supplementary Data 1).

We identified two founder lines that carried the construct and produced *LacZ* staining. The *HoxD*[II1-T-DOM-542] founder line ('allele 542') showed a strong *LacZ* staining in the proximal limb (Fig. 2b, black arrows), with very weak staining appearing in the distal limb from E12.5 onwards (Fig. 2b, white arrows), limited to a small region of the forming digits. Since the II1 transgene was located within T-DOM, which hosts multiple proximal limb enhancers, the strong proximal limb activity likely resulted from the transgene behaving as an enhancer sensor in these cells. The second line identified (*HoxD*[II1-T-DOM-320]) produced a strikingly different staining pattern (Fig. 2c). At E10.5 both the II1 T-DOM 542 and II1 T-DOM 320 lines had strong staining in the proximal limb, but not in the distal limb (Fig. 2b, c; compare white and black arrows). Then, at E11.5, while both lines continued to produce strong proximal limb staining, the 320 line also had strong distal limb staining (Fig. 2c, white arrows). By E12.5 the staining pattern between the lines had almost completely diverged, with II1 T-DOM 542 showing strong proximal and very weak distal staining. In contrast, the II1 T-DOM 320 line produced very strong distal staining but it was nearly absent in the proximal limb (Fig. 2b, c).

To try to understand this difference and to confirm that the II1 transgene was inserted at the expected position, we performed long-read sequencing with the Oxford Nanopore MinION in conjunction with the nCATS protocol[29] to enrich for sequencing reads covering the region around the insertion site (Supplementary Fig. 2). In the 542 samples, we were able to detect a sequencing read that extended the length of the transgene, both homology arms, and several kilobases of the region that flanks the insertion (Supplementary Fig. 2b), thus confirming that the 542 allele was a single copy transgene inserted correctly at the right site. Because this enhancer inserted into T-DOM (hereafter II1 T-DOM) produced the expected allele, we used it for all subsequent experiments. We were not able to completely sequence the II1 T-DOM 320 allele because it contained multiple tandem copies of the insertion, yet we could reconstruct a putative genome based on unique overlapping reads which indicated that a minimum of four copies of the transgene were inserted as a tandem array with multiple orientations (Supplementary Fig. 2c).

From the analysis of these two alleles, we conclude that when inserted as a single copy at the selected position within T-DOM, the distal limb enhancer activity of II1 was almost entirely repressed and thus this sequence behaved there like any other native proximal limb enhancers located in T-DOM[22,24]. Since the heterologous promoter trapped the activity of the surrounding proximal limb enhancers, the final pattern appeared exactly as the reverse pattern for this enhancer when operating from C-DOM or as a randomly integrated transgene. This positive response of the *LacZ* sensor in proximal limb cells also acted as an internal

control; the transgene was capable of transcriptional activity, with a capacity for tissue-specific expression. The repression from the chromatin environment in distal cells was completely alleviated when the transgene was present in multiple tandem copies (Fig. 2c), suggesting a potential micro-structure capable of escaping the negative effect of the local TAD environment. The decreased expression of this allele in proximal cells suggested that the array of transgenes was somehow insulated from receiving the influence of T-DOM proximal enhancers.

**HOX13 transcription factors bind to II1 T-DOM in distal limb cells.** When the II1 transgene was inserted into T-DOM as a single copy, we scored a very low level of *LacZ* staining in E12.5 distal limb cells, detected only at a late stage and appearing as defined spots in the digit mesenchyme, i.e., in cells that do not reflect well the wild type expression pattern which is more widespread at this stage (Figs. 1c and 2b). Because the native II1 enhancer activity seems to be dependent on HOX13 proteins[30], we looked for changes to the accessibility and presence of HOX13 binding to the II1 element within the context of the inactive T-DOM in distal limb cells, to see if HOX13 factors could still access and bind the relocated enhancer sequence.

We first performed an ATAC-seq on embryos homozygous for the II1 T-DOM single-copy insertion along with wild-type controls. We mapped the reads on the mutant genome and used a high mapping quality score (MAPQ30) to ensure that all reads observed on the coverage tracks mapped uniquely to the native II1 element in C-DOM (II1 C-DOM) and the II1 transgene recombined into T-DOM (II1 T-DOM). In wild-type control samples, as expected, no ATAC signal was observed over the native II1 C-DOM, neither in proximal limb bud cells nor in forebrain cells (Fig. 3a, left). In contrast, a strong ATAC signal appeared over the II1 element in distal limb cells (Figs. 1b and 3a; DFL). In the II1 T-DOM transgene, there was a weak signal observed over the *HBB* promoter element in the forebrain and proximal limb cells, the latter signal in proximal forelimb cells (PFL) likely reflecting the *LacZ* staining in response to nearby proximal limb enhancers. In the distal forelimb samples (DFL), there was a further increase in signal over the *HBB* promoter, and the accessible region extended over the II1 enhancer portion of the transgene. This signal over the II1 enhancer was nevertheless weaker than that observed on the II1 element located in its native context (Fig. 3a, ATAC DFL, compare left and right).

The increased accessibility over the enhancer portion of the relocated II1 transgene in T-DOM suggested that HOX13 transcription factor may still be able to bind. We assessed this by performing CUT&RUN experiments using HOXA13 and HOXD13 antibodies on distal limb cells from embryos homozygous for the II1 transgene in T-DOM and wild type controls. To discriminate between reads from II1 in C-DOM or in T-DOM, only those reads containing sequences uniquely mapping to sequences outside the II1 element itself were considered (see methods). As expected, HOX13 proteins bound strongly to the native II1 element in C-DOM, with a peak over the 3′ end of the enhancer in controls (Fig. 3a, left). At the analogous position of the II1 transgene in T-DOM, we observed a similar peak for both HOXA13 and HOXD13, indicating that HOX13 transcription factors were able to bind to the II1 enhancer element when inserted into the T-DOM (Fig. 3a, right). However, the presence of these factors was not able to drive robust transcription of the nearby *LacZ* transgene in distal limb bud cells, even though the promoter was clearly accessible.

To confirm that the HOX13 peaks detected in the II1 enhancer element indeed corresponded to the presence of the expected HOX13 binding motif(s), we performed a motif search analysis[31]

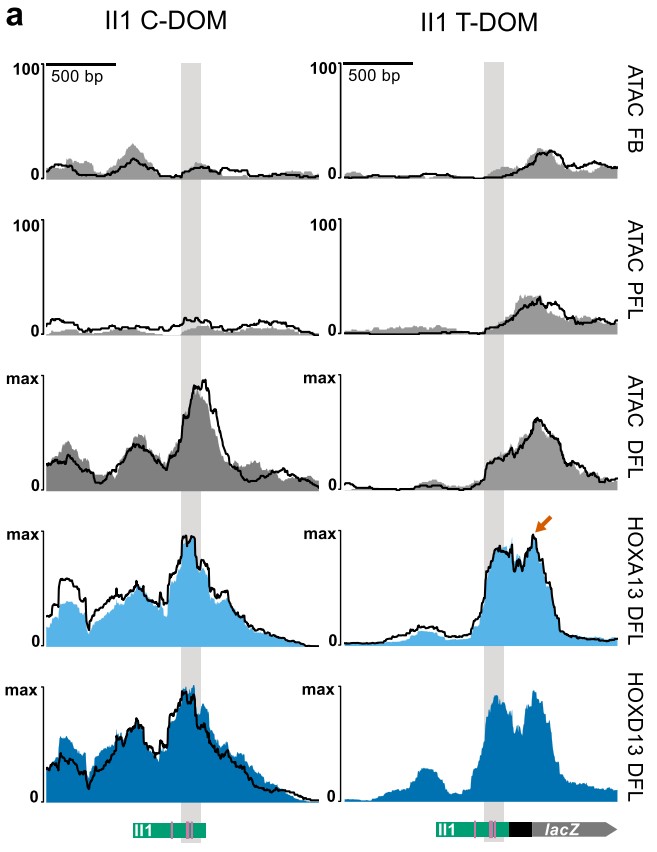

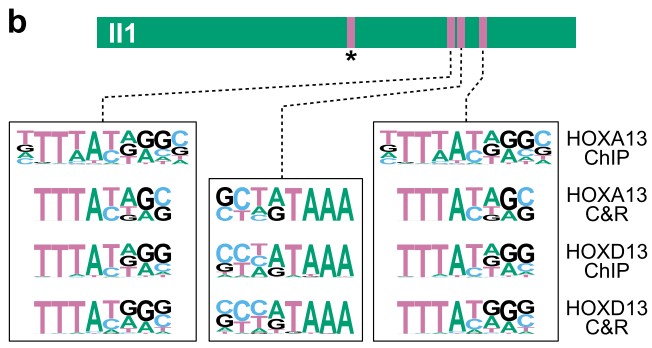

**Fig. 3 HOX13 transcription factor proteins bind to the II1 enhancer in T-DOM. a** ATAC-Seq and CUT&RUN reads mapped to the II1 enhancer sequence, either in its native environment within C-DOM (left column; mm10 chr2:74,074,674-74,076,672), or its targeted recombination site within T-DOM (breeding line 542, right column; mm10 chr2:75,268,925-75,270,923). The II1 C-DOM element is not accessible by ATAC-Seq in E12.5 forebrain (FB), nor in proximal forelimb cells (PFL) samples. At E12.5 it becomes highly accessible in distal forelimb cells (DFL) and is strongly bound by HOX13 proteins. The II1 element in T-DOM has low accessibility in the FB and PFL samples, even though there is a high transcription of the transgene in PFL. Like the II1 element in C-DOM, the II1 enhancer in T-DOM is occupied by HOX13 proteins in distal limb cells. It also shows an additional peak over the *HBB* promoter (orange arrow). This peak is likely a non-specific signal resulting from promiscuous MNase activity used in the CUT&RUN technique. In all samples but HOXD13 in the II1 T-DOM allele, experiments were performed in duplicate. One biological replicate is plotted as a solid color and the other is shown as a superimposed black line (*n* = 2). Green rectangles below indicate the position of the II1 enhancer element relative to the peak signal; the position of the four HOX13 binding sites is indicated by pink lines. **b** On top is a schematic of the II1 enhancer element with the four HOX13 motifs indicated as pink bars. The pink bar with an asterisk indicates the motif position that is not near the HOX13 and ATAC peaks. At the bottom are the three HOX13 motifs identified by HOMER motif discovery in the CUT&RUN experiments here and the E11.5 whole forelimb ChIP-seq[25].

staining in some specific distal limb cells, even though this staining was distinct from the strong and general accumulation scored with the randomly integrated transgene (II1 TgN, compare Figs. 1c and 2b). Instead, it could be caused by some other factor(s) taking advantage of the pioneer activity of HOX13 factors[18,30]. We verified this by testing the necessity of these three HOX13 binding sites in the II1 enhancer recombined into T-DOM. We implemented a CRISPR approach in vivo to delete the HOX13 binding sites in the II1 T-DOM transgene, by using guides to delete either two or three of the HOX13 binding sites (Fig. 4a). Embryos hemizygous for the II1 enhancer in T-DOM were electroporated with CRISPR guides and Cas9 protein and the embryos were collected at E12.5 and stained for *LacZ*. Subsequently, the induced mutations were confirmed by Sanger sequencing (Supplementary Data 2). While the *LacZ* staining in proximal limb cells was not affected by these mutations, in all cases where the binding sites were deleted, the remaining *LacZ* staining found in distal limb cells of II1 T-DOM transgenic embryos was completely ablated even after overstaining the samples (Fig. 4b, Del TFBS, and Supplementary Fig. 4-1a).

Of note, in several F0 embryos, we observed very strong proximal and distal limb *LacZ* expression (Fig. 4b and Supplementary Fig. 4-1b, Del C-T). When we sequenced the genomic DNA of these embryos, we found that they all contained a large deletion extending from the native II1 enhancer element within C-DOM up to the II1 transgene within the T-DOM region (approximately 1.2 Mb in length), due to the use of guide target sequences present in both copies of the II1 sequence (scheme in Supplementary Fig. 4-1c). In these embryos, the HOX13 binding sites are deleted and the *LacZ* transgene, formerly located within the T-DOM, was fused with a portion of the C-DOM (Supplementary Fig. 4-1c). In such a configuration, it is very likely that the enhancer remaining 5′ to the II1 site in C-DOM (island I) was then able to act on the transgene driving expression in the distal limb (Fig. 4b and Supplementary Fig. 4-1c), a regulatory influence obviously not permitted in the presence of an integral native T-DOM. *LacZ* expression in proximal cells was controlled by those remaining proximal enhancers located

for our CUT&RUN samples and for a previously reported dataset using ChIP-Seq[25]. In both datasets, we found motifs for HOX13 factors within the II1 element at four different positions. Three of these sites are closely clustered at the 3′ end of the enhancer element (Fig. 3b, pink bars) and match the peak summit for HOXA13 and HOXD13 in both the native II1 enhancer environment in C-DOM and in the transgene in T-DOM (Fig. 3a, gray columns). An additional motif was found within the II1 region but locates outside the peak region (Fig. 3b, asterisk). The three clustered motifs match previously reported HOX13 motifs[25,30] and their position in relation to the position of CUT&RUN reads suggested that these sites are, in large part, responsible for the distal limb enhancer activity of the native II1 element in C-DOM.

**HOX13 binding sites are essential to II1 enhancer activity in distal limb buds**. The presence of HOX13 factors bound to the II1 enhancer sequence when integrated into T-DOM suggested that they may be responsible for the weak remaining *LacZ*

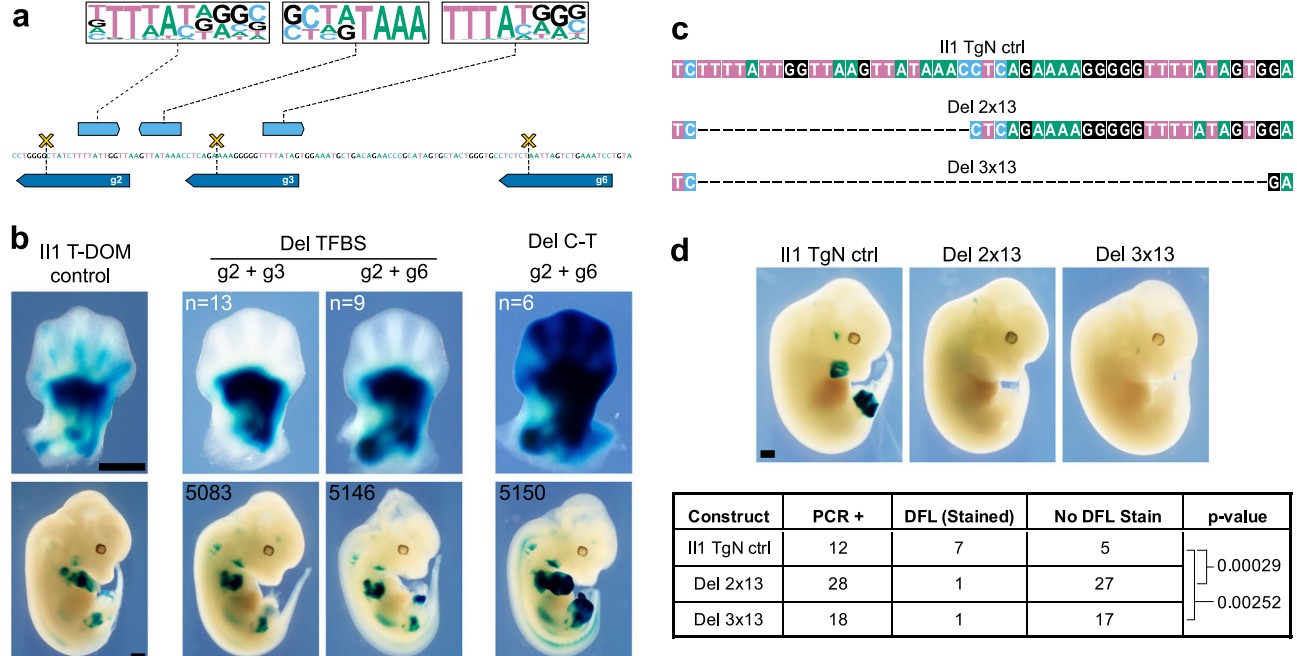

**Fig. 4 The effects of deleting HOX13 binding sites. a** On top are the HOX13 motifs in the Il1 element, identified in Fig. 3b, with their positions indicated below (light blue boxes with orientations). The dark blue boxes below the DNA sequence indicate the positions and orientations of the CRISPR guides. Combinations of guides were used to generate small deletions (g2 + g3 or g2 + g6). The yellow crosses indicate the approximate cutting position of the Cas9. **b** E12.5 F0 *LacZ* stained embryos containing these deletions. The left panel is a Il1 T-DOM control limb (no CRISPR cutting), with weak staining in the distal digit mesenchyme (dozens were stained). The two central panels are representative embryos (sample size n indicated in upper left corners) with the indicated deletions (see Supplementary Fig. 4-1 for images of all embryos including these two #5083 and #5146). Embryos carrying either the g2 + g3 or g2 + g6 deletions lost all staining in distal limb cells. In the same litters, we observed some embryos with strong distal limb staining (as shown in the right panel, see also Supplementary Fig. 4-1b, c). They contained a deletion that extends from the native Il1 element in C-DOM to the Il1 transgenic element in T-DOM, due to the guide RNA sequence presence at both sites (scheme in Supplementary Fig. 4-1c). Deletions in all embryos were sequenced (Supplementary Data 2). Embryos with ambiguous sequencing results or mosaicism were not used in this analysis. **c** DNA sequence, mapping to the putative HOX13 binding sites in the Il1 enhancer elements (see **a**). Il1 transgenic embryos were generated using these variants as the transgenic material. The top track is the wild-type sequence used in control embryos (Il1 TgN ctrl). The Del 2 × 13 sequence lacks the two centromeric motifs (see **a**), while the Del 3 × 13 sequence lacks all three motifs for putative HOX13 binding. **d** On top are *LacZ* stained E12.5 transgenic embryos, the table below reports the number of embryos positive for the transgene by PCR (PCR + ), the second column is the number of embryos that stained in the distal forelimb (DFL), followed by the embryos with no staining in the DFL (No DFL Stain). The p values are determined by two-sided Fisher's exact tests. Representative embryos are shown for each transgene. See also Supplementary Fig. 4-2. Scale bars are 0.5 mm.

telomeric to the deletion breakpoint (Fig. 2a and Supplementary Fig. 4-1c, CS65 and PLEs).

As a control for the requirement of HOX13 factors for the function of the Il1 enhancer sequence, we generated two variants of the Il1 TgN transgene construct lacking either two (Del 2 × 13) or three (Del 3 × 13) HOX13 binding sites (Fig. 4c and Supplementary Data 1). We then injected these constructs into embryos to produce random integration events and then stained for *LacZ*. In nearly all cases, neither the Del 2 × 13, nor the Del 3 × 13 variants were able to produce distal limb staining, when compared with the embryos containing the complete Il1 enhancer sequence (Fig. 4d). There were two exceptions to this: in one Del 2 × 13 embryo, we observed a clear distal limb staining although the proximal boundaries were very different than that seen in the normal Il1 transgene (Supplementary Fig. 4-2, Del 2 × 13, yellow asterisk) and, in one Del 3 × 13 embryo, there was strong proximal limb staining and a portion of this staining extended into the posterior portion of the distal limb (approximately digit 5), yet most of the distal limb staining was absent (Supplementary Fig. 4-2, Del 3 × 13, yellow asterisk). In the remaining embryos carrying the transgene, twenty-seven embryos with the Del 2 × 13 transgene and seventeen embryos with the Del 3 × 13 transgene did not produce any distal limb staining (Fig. 4d and Supplementary Fig. 4-2).

As a final control experiment that unifies the two experiments above, we targeted the insertion of two additional variants of the Il1 transgene into the same position of T-DOM as the Il1 T-DOM 542 allele. In the first variant, we used the Del 3 × 13 transgene construction used to test the need for the three HOX13 binding sites when integrated at random locations throughout the genome (Fig. 4c–d). When we stained embryos carrying this transgene in the T-DOM, we again observed very strong proximal limb staining that matched the 542 allele, indicating that the transgene behaved as a T-DOM enhancer sensor in proximal limb cells (Supplementary Fig. 4-3a), but there was no staining in the distal limb. In the second variant, we completely removed the Il1 enhancer element from the transgene and inserted into the T-DOM. In this construction the *LacZ* staining pattern also produced strong proximal limb staining and no staining in the distal limb (Supplementary Fig. 4-3b).

Altogether, these results indicate that even in the presence of bound HOX13 proteins, which are normally the essential factors for its activation, the Il1 enhancer sequence recombined within T-DOM was unable to express its full potential. Indeed, only a weak remnant of a transcriptional activity was scored in distal cells, at a late stage and low level, even though the *LacZ* sensor system could work at high efficiency in proximal cells. This

suggested that, in distal limb cells, the surrounding chromatin context of T-DOM exerted a dominant effect to repress the activity that this sequence normally displays when positioned within C-DOM, even if the binding of HOX13 factors was still observed.

**A TAD-driven repression of distal enhancers in proximal limb cells?** In order to challenge this negative in-*cis* effect, we used CRISPR to delete portions of the T-DOM adjacent to the introduced enhancer transgene and then evaluated the effect of these deletions on transcription of the *LacZ* sensor construct. Fertilized eggs hemizygous for the II1 T-DOM recombined enhancer allele were electroporated with CRISPR guides and Cas9 protein (Supplementary Data 1). Since the DNA sequences targeted by guides are present on both the wild-type and the II1 T-DOM alleles (Fig. 5a and Supplementary Data 1), we used genotyping PCR to screen for embryos carrying the expected deletion and used changes in *LacZ* staining to associate the deletion with the II1 T-DOM chromosome. As a control, we used littermate embryos that contained the same deletion but on the wild-type (non-transgenic) chromosome.

The first deletion extended from the 3′ end of the *Mtx2* gene, up to, but not including, the II1 enhancer element in the T-DOM (Fig. 5a; Del *Mtx2*-II1-T-DOM). This deleted region of T-DOM contains the CS39 and CS93 proximal limb enhancers[22,32]. In this deletion, we only scored a slight reduction in the extent of the *LacZ* staining in proximal limb cells (Fig. 5b, black arrows), likely due to the removal of some proximal limb enhancers. In distal limb cells, there was a clear increase in the *LacZ* staining throughout the distal limb mesenchyme spanning almost the entire digital plate, as compared to the same deletion on the wild-type chromosome (Fig. 5b, compare left and right, Supplementary Fig. 5a).

The second deletion extended from the 3′ end of the *LacZ* transgene to the telomeric end of the T-DOM regulatory landscape (Fig. 5a, Del II1-T-DOM-*Hnrnpa3*). This portion of T-DOM contains the CS65 and PLE proximal limb enhancer elements[22,26]. In this deletion, we observed a severe loss of *LacZ* staining in proximal limb cells (Fig. 5c and Supplementary Fig. 5b). In distal limb cells, there was a slight difference in the distribution of *LacZ* positive cells, yet no obvious increase in staining when compared to the control chromosome, unlike in the former deletion (Fig. 5c and Supplementary Fig. 5b). These results showed that staining could be recovered when the centromeric flanking piece of T-DOM was removed and hence that this chromatin segment somehow exerted a robust repressive effect on the II1 transgene in distal limb bud cells.

**The II1 T-DOM transgene contacts the *HoxD* gene cluster.** In distal limb bud cells at E12.5, strong chromatin contacts are detected between the II1 enhancer sequence (or a larger sequence including it) and the "posterior" part of the *HoxD* cluster[17]. Along with other C-DOM cis-regulatory regions, the II1-Hox cluster interactions collectively sustain activation of *Hoxd13* to *Hoxd11* in the digital plate[17]. Because of this, we wondered how this enhancer sequence would behave when relocated within the 3D chromatin space of the neighboring TAD. In other words, would it maintain its contacts with these 'distal' limb genes (*Hoxd13* to *Hoxd11*), not establish any contacts with the cluster at all, or would it adopt the interaction tropism of its new T-DOM neighborhood for the more 'proximal' limb genes (*Hoxd10*, *Hoxd9*, *Hoxd8*)? We performed Capture Hi-C (CHi-C) on E12.5 proximal and distal forelimb cells micro-dissected either from wild-type embryos, or from embryos homozygous for the

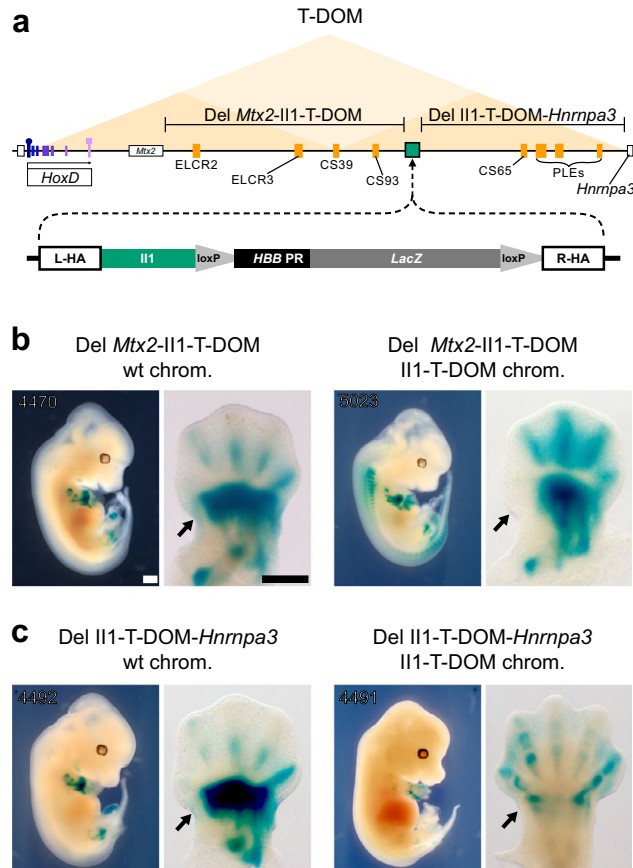

**Fig. 5 Restoring distal enhancer activity by removing the T-DOM chromatin environment. a** Schematic of T-DOM with the *Hoxd* gene cluster on the left (purple boxes), *Hoxd1* with a square and *Hoxd13* with a circle on top. The *Mtx2* gene is next to *Hoxd1* and *Hnrnpa3* is the small white box with black border on the right of T-DOM. The position of the II1 transgene insertion into T-DOM is indicated by a green rectangle with black border (mm10 chr2:75269597-75269616). Orange rectangles are known proximal limb enhancers. The two regions deleted by CRISPR are indicated above the genomic map (Del *Mtx2*-II1-T-DOM and Del II1-T-DOM-*Hnrnpa3*). **b** The effect on *LacZ* staining caused by deleting the centromeric portion of T-DOM (Del *Mtx2*-II1-T-DOM). The left two panels are a control embryo with the deletion on the wild-type chromosome, and so are not expected to show changes in staining. In the right two panels, embryos with the deletion on the same chromosome as the II1 T-DOM 542 transgene, showing a light loss of staining in the proximal domain (black arrows) and a gain in the distal domain. All embryos used were photographed and included in Supplementary Fig. 5a, $n = 6$ and $n = 5$, respectively. **c** The effect of deleting the telomeric portion of T-DOM (Del II1-T-DOM-*Hnrnpa3*) on *LacZ* staining, with an almost complete loss of staining in the proximal domain (black arrows) and no substantial impact on the distal domain ($n = 5$, and $n = 6$, respectively). The embryos shown here are also displayed in Supplementary Fig. 5b to show the complete series of stained embryos. Scale bars are 0.5 mm.

recombined II1 T-DOM transgene. The captured reads were mapped onto the mutant genome excluding all reads that would ambiguously map to both the II1 enhancer in C-DOM and in T-DOM, i.e., sequence reads that would not extend outside of the enhancer itself and hence could not be uniquely assigned to either one of the two sites.

In both the proximal and distal forelimb datasets, strong contacts were established between the II1 enhancer element within T-DOM and the *HoxD* gene cluster, as revealed by the

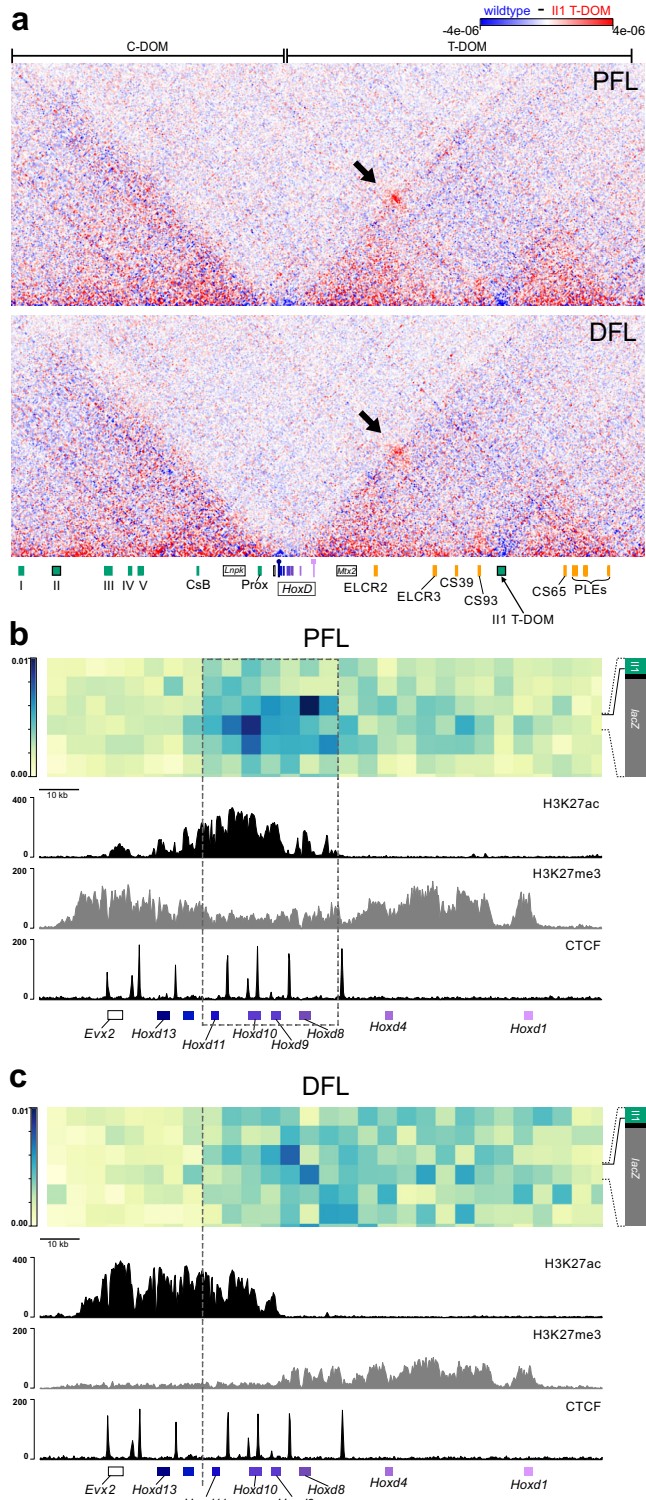

**Fig. 6 The Il1 enhancer in T-DOM contacts the *Hoxd* gene cluster.**
**a** Capture Hi-C maps at 5 kb bin resolution over the entire *HoxD* locus (mm10 chr2:73,950,000–75,655,000), displayed as the subtraction of wild type signal (blue) from the Il1 T-DOM 542 allele signal (red). The Il1 enhancer transgene inserted in the T-DOM (arrow on the green rectangle at the bottom) produces clear contacts with the *Hoxd* gene cluster (black arrow pointing to the red bins). The contacts are established in both proximal forelimb (PFL) and distal forelimb (DFL) cells. **b** Contacts between the *Hoxd* gene cluster (x-axis, genes are indicated below) and the region covering the Il1 T-DOM reporter transgene (y-axis) in proximal forelimb cells (PFL). The Il1 T-DOM construct is schematized on the y-axis for clarity (right), a solid black line indicates where the bin boundary aligns to the transgene. The Il1 enhancer is shown in green, the *HBB* promoter is black, and the *LacZ* gene is in gray. The panels below are the H3K27ac, H3K27me3[32], and CTCF[23] ChIP-Seq tracks from wild type PFL cells, aligned with the interaction matrix above. The strongest contacts are between the Il1 T-DOM reporter transgene and the region around *Hoxd8* and *Hoxd10*, matching genes transcribed in proximal limb cells, surrounded by a gray dashed box. **c** Same as in **b** but using distal forelimb cells (DFL)[23,32]. The inserted Il1 T-DOM transgene establishes more diffuse contacts, extending from *Hoxd1* to *Hoxd11* and stopping abruptly before those genes highly that are expressed in distal cells (*Hoxd12*, *Hoxd13*).

These alignments revealed clear differences between contacts in proximal and distal cells, in both their relative strengths and localization. In proximal cells, the Il1 enhancer and the *HBB* promoter formed contacts mostly concentrated over the *Hoxd8* to *Hoxd10* region (Fig. 6b, gray dashed box). However, even at this 5 kb resolution, we were not able to resolve if the contacts were being mediated by the Il1 transcription factor binding sites or the *HBB* promoter, because both portions of the transgene are within the same *DpnII* restriction fragment (Fig. 6b, c). In distal limb bud cells, the contact dynamic was quite different. While the fragment containing the Il1 enhancer and the *HBB* promoter continued to form the strongest contacts with the cluster, the region of highest contact had shifted from the 3′ end of *Hoxd8* to the 5′ end, and the robust contacts detected around *Hoxd10* in proximal cells had nearly disappeared (Fig. 6c). Overall, the contacts were more evenly distributed over the region extending from *Hoxd1* to *Hoxd9*, but they were excluded from the *Hoxd12* to *Hoxd13* region (Fig. 6c, gray dashed line).

The contacts between the Il1 T-DOM transgene and the *HoxD* cluster changed between proximal and distal limb bud cells, exactly matching the distribution of active *versus* inactive chromatin in these two developmental contexts, respectively. In proximal limbs, the region of enriched contacts corresponded to a depletion in H3K27me3 marks and an enrichment in H3K27ac, which exactly matches the region of *HoxD* that is actively transcribed in proximal limb cells[22,33]. In distal limb cells, the contacts became more evenly distributed and correlated with the presence of H3K27me3, stopping abruptly within the H3K27ac-positive portion of *HoxD* cluster, before reaching *Hoxd13* and *Hoxd12* (Fig. 6c). Therefore, the Il1 enhancer sequence, when recombined into T-DOM followed the spatial distribution of T-DOM proximal limb enhancers (Supplementary Fig. 6), even though it had no intrinsic proximal limb enhancer activity, as demonstrated by random transgenesis (Fig. 1c, Supplementary Fig. 1-1b, and Fig. 4d). Regardless of the context and the developmental time, when positioned into T-DOM, it never contacted the *Hoxd12* to *Hoxd13* region, which is the part of the cluster that this enhancer sequence normally contacts with the highest affinity in distal limb bud cells, nor did it influence in any way or in any cell type the chromatin structure that is normally found in T-DOM.

subtraction of the mutant contact signal from the wild type (Fig. 6a, black arrows; Supplementary Fig. 6, black arrows). This gain of contacts between the recombined Il1 enhancer and the *HoxD* cluster in both proximal and distal cells, were the only noticeable change induced by the presence of the transgene on the general chromatin configuration of the locus (Fig. 6a). In order to evaluate these new contacts in greater detail, we plotted pairwise heatmaps between the *HoxD* cluster and Il1 T-DOM, as well as H3K27me3, H3K27ac, and CTCF ChIP-seq datasets[23,32].

## Discussion

The importance and status of enhancer sequences have evolved considerably since their discovery[1]. Initially described as short non-coding sequences that can increase the transcription rate of a target gene at a distance and regardless of orientation, they are now known to modulate gene expression in many different ways[34]. Enhancers have particular importance for genes with specialized expression patterns, producing transcription in specific cell types and tissues, at precise times and quantities, either during development or subsequently[35]. These sequences are thought to have evolved along with the emergence of novel body structures, recruiting genes already functional elsewhere, to accompany or trigger the formation of these novelties[7]. In vertebrates, where multi-functionality is common for genes having important functions during development, enhancers have accumulated in the vicinity of transcription units, forming regulatory landscapes[3] that sometimes extend over several megabases (see[2]). Within these landscapes, gene activation potential can be distributed across multiple enhancers[17,36] often leading to functional redundancy between them or to more complex interactions[18,37]. Alternatively, enhancer sequences can be grouped together in a more compact manner, either to ensure a coordinated function, as exemplified by the *Globin* genes[38–40], or to maximize transcription in a given cellular context such as the compact regulatory structure referred to as super enhancers[41,42].

Enhancers are often embedded within TADs, which are regions where certain DNA–DNA interactions are favored while adjacent regions are excluded from the interaction space[4–6]. As illustrated at the *HoxD* locus, the genomic dimensions of TADs sometimes correspond to the extents of regulatory landscapes[22], with TADs providing boundaries to the interaction space of enhancers within the three-dimensional organization of the genome. Consequently, TADs have been considered to be permissive structures augmenting enhancer and promoter interactions, while simultaneously providing borders that prevent enhancers from interacting with elements outside the TAD, and hence to regulate genes in an inappropriate manner[43,44]. In this view, however, the enhancer sequence is considered as a regulatory element that can explore and act within the nuclear space somewhat freely unless TAD borders are present to frame its realm of action. Alternatively, there may be some loci where the function of an enhancer sequence can be subordinated to the global chromatin context of a given TAD thus introducing a level of regulatory control derived from a large chromatin domain rather than by individual DNA sequences[22,36,45].

The transfer of enhancer II1 into T-DOM may illustrate such a case, where a potent and highly penetrant enhancer sequence is inhibited precisely in those cells where it normally functions, by placing it within a global environment that is not operational in these cells. In its normal location among C-DOM enhancers, the II1 sequence is bound by HOX13 factors[24,25,30], which are responsible for its strong activation potential as demonstrated here by the deletion of these sites in the transgenic context, which leads to the loss of *LacZ* staining in digit cells. When recombined into T-DOM, the II1 enhancer is silenced yet it still recruits HOX13 factors and hence its silencing cannot be attributed to the absence of the necessary activating factors. In fact, HOX13 factors are also bound to T-DOM whenever this TAD becomes inactive in distal limb cells and covered by H3K27me3 marks[24], suggesting that the same factors may act in both a positive and a negative manner in different chromatin contexts.

One potential explanation for this observation is that HOX13 factors may function with more than one modality, depending on their context. On the one hand, these factors may bind to and activate an enhancer-reporter transgene in a sequence-specific manner, as many transcription factors do, leading to the pattern described herein and its absence in transgenes lacking the binding sites. On the other hand, both HOXD13 and HOXA13 proteins contain a large poly-alanine stretch[46], which may potentially drive the formation of phase-separated globules by co-condensing with transcriptional co-activators/co-repressors, as shown for these and other genes[38,47]. It is thus possible that, due to their high content in bound HOX13 proteins, both C-DOM and T-DOM are used to form large transcription-hub condensates[18,38], leading to a positive transcriptional outcome for C-DOM in distal limb cells, and a negative outcome for T-DOM, within the same cells, due to the inclusion of additional co-factors that are specific to each domain[48]. This latter explanatory framework would also account for why the deletion of the enhancer II1 in vivo had essentially no detectable effect upon transcription of target *Hoxd* genes in distal limbs, much like what was reported for C-DOM enhancers used for external genitals[18] as well as in other comparable instances[37]. In both cases, removing a single component of the aggregate would not matter too much, whereas removing several related enhancers would then have a measurable impact.

This situation contrasts with those where a single enhancer is responsible for target gene activation, the deletion of which usually seriously impairs the structure (see e.g[49,50].). However, the former mechanism would not preclude the capacity for a single distal limb enhancer to trigger *LacZ* expression, as illustrated either by the weak staining detected when II1 was relocated into T-DOM, or when a large deletion brought the *LacZ* reporter close to Island 1, following the fusion between parts of C-DOM and T-DOM.

The II1 enhancer was selected because it is one of the strongest and most penetrant distal limb cell enhancers reported thus far[27], yet it was silenced when introduced into T-DOM. A related observation has been made for several native T-DOM limb enhancers, which have a strong proximal specificity when inside T-DOM, while they can work efficiently in distal cells as well when randomly integrated as transgenes[24]. The repression of T-DOM in distal limb cells is reflected by the presence of large arrays of H3K27me3 marks[22], and it is thus possible that the recombined II1 enhancer was included in this negative chromatin domain. The analysis of the II1 T-DOM 320 line (Fig. 2c) suggests that this repression can be competed out by the presence of multiple copies of the enhancer-reporter construct, perhaps due to the formation of a particular sub-structure escaping the negative effect of T-DOM, or simply because of the accumulation of some transcription factors. While this line was not analyzed further due to technical difficulties associated with duplicated genomic sequences, it indicated that special attention should be given to copy number when interpreting results from transgenic experiments.

The negative effect of T-DOM over the II1 enhancer construct in-*cis* was further suggested by our flanking deletions analyses. The (Del II1-T-DOM-*Hnrnpa3*) telomeric deletion did not substantially change the weak distal staining of the transgene, yet it severely reduced expression in the proximal domain, showing that the most potent proximal enhancers were located in the deleted interval. In contrast, the (Del *Mtx2*-II1-T-DOM) centromeric deletion consistently had the opposite effect, with only a slight reduction of the activity in the proximal limb domain while distal expression was re-activated, thus mapping the main T-DOM region carrying the repressive effect between the gene cluster and the integration site of the enhancer-reporter construct. However, it was not assessed whether this reactivated distal *LacZ* staining is due to the II1 enhancer itself or to the de-repression of other T-DOM enhancers, which would then act on the *HBB* promoter due to the in-*cis* proximity.

Finally, the recombined II1 construct was able to specifically and rather strongly contact the *HoxD* cluster, in both proximal (when T-DOM is active) and distal (when T-DOM is inactive) limb bud cells. In these two instances, however, the contacts were distributed differently. In proximal cells, contacts were established with the specific part of the cluster that is heavily transcribed, following the behavior of other T-DOM located enhancers[22]. In this case, these interactions were not directly driven by CTCF, for the integration site was selected at a distance from such sites. The enhancer was likely included in a global structure interacting with specific *Hoxd* promoters and possibly supported by CTCF sites within both the gene cluster and T-DOM[51]. In distal cells, the interactions were less specific and likely reflected contacts between large H3K27me3 decorated chromatin segments[52,53], as suggested by the absence of contacts with those genes heavily transcribed in distal cells. There again, the interaction profile of the II1 enhancer followed its new neighboring proximal enhancers.

Altogether, we conclude that in distal limb bud cells, the necessary decommissioning of all previously acting proximal enhancers is partly achieved -or secured- by a TAD-wide silencing mechanism, as illustrated by the appearance of large domains of H3K27me3[22,24]. This mechanism is potent enough to prevent the expression of one of the strongest distal limb enhancers, after its recombination within T-DOM. This silencing is partly alleviated when a large piece of flanking chromatin is removed, suggesting the importance of neighboring sequences within the TAD to achieve this effect. When this distal enhancer was introduced into the proximal limb TAD, it followed the interaction tropism of its new neighbor proximal limb regulatory landscape, yet could not escape its repressive effects in the distal limb, thus illustrating the potential of chromatin domains, in some cases, to impose another level of coordinated regulation on top of enhancer sequence specificities.

## Methods

**Animal work**. All experiments were approved and performed in compliance with the Swiss Law on Animal Protection (LPA) under license numbers GE 81/14 and VD2306.2 (to D.D.). All animals were kept in a continuous back cross with C57BL6 × CBA F1 hybrids. The sex of the embryos was not considered in this study. Mice were housed at the University of Geneva Sciences III animal colony with light cycles between 07:00 and 19:00 in the summer and 06:00 and 18:00 in winter, with ambient temperatures maintained between 22 and 23 °C and 45 and 55% humidity, the air was renewed 17 times per hour.

**Genotyping**. When samples were to be used directly for experiments, a rapid protocol was implemented: Yolk sacs were collected and placed into 1.5 ml tubes containing Rapid Digestion Buffer (10 mM EDTA pH8.0 and 0.1 mM NaOH) then placed in a thermomixer at 95 °C for 10 min with shaking at 900 rpm. While the yolk sacs were incubating, the PCR master mix was prepared with Z-Taq (Takara R006B) (see Supplementary Data 1 for genotyping primers) and aliquoted into PCR tubes. The tubes containing lysed yolk sacs were then placed on ice to cool briefly and quickly centrifuged at high speed. The lysate (1 μl) was placed into the reaction tubes and cycled 32× (2 s at 98 °C, 2 s at 55 °C, 15 s at 72 °C). Twenty microliters of the PCR reaction were loaded onto a 1.5% agarose gel and electrophoresis was run at 120 V for 10 min. Alternatively, when samples could be kept for some time, a more conventional genotyping protocol was applied; Tail Digestion Buffer (10 mM Tris pH8.0, 25 mM EDTA pH8.0, 100 mM NaCl, 0.5% SDS) was added to each yolk sac or tail clipping at 250 μl each along with 4 μl Proteinase K at 20 mg/ml (EuroBio GEXPRK01-15) and incubated overnight at 55 °C. The samples were then incubated at 95 °C for 15 min to inactivate the Proteinase K and stored at −20 °C until ready for genotyping. Genotyping primers (Supplementary Data 1) were combined with Taq polymerase (Prospec ENZ-308) in 25 μl reactions and cycled 2× with $T_a = 64$ °C and then cycled 32× with $T_a = 62$ °C.

***LacZ* staining**. Embryos were collected in ice-cold 1× PBS in a 12-well plate. They were then fixed for 5 min at room temperature in freshly prepared 4% PFA with gentle shaking on a rocker plate. After fixing they were washed three times in washing solution (2 mM MgCl₂, 0.01% Sodium Deoxycholate, 0.02% Nonidet P40, and 1× PBS) for 20 min each at room temperature on a rocker plate. After

approximately 1 h of washing, the wash solution was removed and replaced with staining solution (5 mM Potassium Ferricynide, 5 mM Potassium Ferrocynide, 2 mM MgCl2 hexahydrate, 0.01% Sodium Deoxycholate, 0.02% Nonidet P40, 1 mg/ml X-Gal, and 1× PBS). The plate was wrapped in aluminum foil and placed on a rocker plate overnight at room temperature. The following morning the staining solution was removed and the embryos were washed three times in 1× PBS and then fixed in 4% PFA for long-term storage. Images of embryos were collected with an Olympus DP74 camera mounted on an Olympus MVX10 microscope using the Olympus cellSens Standard 2.1 software.

**Whole-mount in situ hybridization (WISH)**. Embryos were collected at E12.5 and processed following a previously reported WISH procedure[54]. Briefly, embryos were fixed overnight in 4% PFA at 4 °C. The following day they were washed and dehydrated through 3 washes in 100% methanol and then stored at −20 °C until ready for processing. Each sample was prepared with Proteinase K (EuroBio GEXPRK01-15) at 1:1000 for 10 min. Hybridizations were performed at 69 °C and post-hybridization washes were performed at 65 °C. Staining was performed with BM-Purple (Roche 11442074001). All WISH were performed on at least three biological replicates. Images of embryos were collected with an Olympus DP74 camera mounted on an Olympus MVX10 microscope using the Olympus cellSens Standard 2.1 software.

**RT-qPCR**. Embryos were isolated from the uterus and placed into 1× DEPC-PBS on ice. The yolk sacs were collected for genotyping. The embryos were transferred into fresh 1× DEPC-PBS and the distal limb portion was excised, placed into RNALater (ThermoFisher AM7020), and stored at −80 °C until processing. Batches of samples were processed in parallel to collect RNA with Qiagen RNEasy extraction kits (Qiagen 74034). After isolating total RNA, first strand cDNA was produced with SuperScript III VILO (ThermoFischer 11754-050) using approximately 500 ng of total RNA input. cDNA was amplified with Promega GoTaq 2× SYBR Mix and quantified on a BioRad CFX96 Real Time System. Expression levels were determined by dCt (GOI–Tbp) and normalized to one for each condition by dividing each dCT by the mean dCT for each wild-type set. Supplementary Data 1 contains the primer sequences used for quantification. Box plots for expression changes and two-tailed unequal variance *t* tests were produced in DataGraph 4.6.1.

**CUT&RUN**. Embryos were collected in ice-cold 1× PBS and yolk sacs were processed according to the rapid genotyping protocol described above. Embryos with the correct genotype were transferred to fresh PBS and dissected. The dissected tissue samples were transferred into 1× PBS containing 10% FCS and then digested with collagenase (see ATAC-Seq protocol below). For the HOXD13 and HOXA13 CUT&RUN, pools of cells from individual embryos were processed. All samples were processed according to the CUT&RUN protocol[55] using a final concentration of 0.02% digitonin (Apollo APOBID3301). Cells were incubated with 0.5 μg/100 μl of anti-HOXD13 antibody (Abcam ab19866), 0.5 μg/100 μl of anti-HOXA13 (Abcam Ab106503) in Digitonin Wash Buffer at 4 °C. The pA-MNase was kindly provided by the Henikoff lab (Batch #6) and added at 0.5 μl/100 μl in Digitonin Wash Buffer. Cells were digested in Low Calcium Buffer and released for 30 min at 37 °C. Sequencing libraries were prepared with KAPA HyperPrep reagents (07962347001) with 2.5ul of adapters at 0.3uM and ligated for 1 h at 20 °C. The DNA was amplified for 14 cycles. Post-amplified DNA was cleaned and size selected using 1:1 ratio of DNA:Ampure SPRI beads (A63881) followed by an additional 1:1 wash and size selection with HXB. HXB is equal parts 40% PEG8000 (Fisher FIBBP233) and 5 M NaCl.

CUT and RUN libraries were sequenced paired-end on a HiSeq4000 sequencer and processed as in[26], mapped either on mm10 or on the II1TDOM-542 mutant genome. The E11.5 whole-forelimb HOXA13 and HOXD13 ChIP-Seq datasets (SRR3498934 of GSM2151013 and SRR3498935 of GSM2151014) as well as E12.5 distal and proximal forelimb H3K27Ac and CTCF ChIP-Seq datasets (SRR5855214 of GSM2713703, SRR5855215 of GSM2713704, SRR5855220 of GSM2713707, and SRR5855221 of GSM2713708) were processed similarly to what has been previously published[56]. Adapter sequences and bad quality bases were removed with Cutadapt[57] version 1.16 with options -a GATCGGAAGAGCACACGTC TGAACTCCAGTCAC -A GATCGGAAGAGCGTCGTGTAGGGAAAGAGT GTAGATCTCGGTGGTCGCCGTATCATT -q 30 -m 15 (−A being used only in PE data sets). Reads were mapped with bowtie[58] 2.4.1 with default parameters on mm10. Alignments with a mapping quality below 30, as well as discordant pairs for PE datasets, were discarded with samtools view version 1.8[59,60]. Coverage and peak calling were computed by macs2[61] version 2.1.1.20160309 with options --bdg --call-summits --gsize '1870000000', and -f BAMPE for PE. The HOX13 motifs where identified with findMotifsGenome.pl from the Homer tool suite[31] using the narrowPeak of HOXA13 and HOXD13 ChIP as well as the third replicate of the HOXA13 and HOXD13 CUT and RUN with the option -size 50. The best motif of each of these four datasets was used to scan the sequence of the II1 enhancer. Four motifs were identified, for the three displayed in Fig. 3b, the logo of the motif giving the best score is shown.

**ATAC-Seq**. Samples used for ATAC-Seq were processed following the original protocol. Cells were collected in 1× PBS on ice and yolk sacs were collected for each sample. Embryos were rapidly genotyped (see above) and those with the correct

genotype were transferred to fresh 1× PBS and dissected. Tissue samples were transferred into 300ul 1× PBS containing 10% FCS on ice until ready for processing. To each sample, 8ul of collagenase (at 50 mg/ml, Sigma C9697) was added and tubes were placed in a Thermomixer at 37 °C with shaking at 900 rpm for approximately 5 min or until the samples were completely disaggregated. The samples were then placed into a centrifuge at 4 °C and centrifuged at $500 \times g$ for 5 min. The supernatant was removed and cells were gently resuspended in ice-cold 1× PBS. The cells in each sample were counted with a Countess (ThermoFisher) using Trypan Blue and checked for viability >90%. The volume needed to contain 50,000 cells was determined and that volume was transferred to a new tube and centrifuged at 4 °C at $500 \times g$ for 5 min. The supernatant was removed and the cell pellet was gently resuspended in lysis buffer then immediately centrifuged at $500 \times g$ for 10 min at 4 °C. The lysis buffer was removed and the cell pellet was gently resuspended in 50 µl tagmentation mix (Nextera FC-121-1030) and then incubated at 37° in a thermomixer at 300 rpm for 30 min. The samples were then mixed with Buffer QG from the Qiagen MinElute PCR Purification kit (28004) and processed according to that protocol, and then eluted from the column in 11ul EB. Samples were stored at −20 °C until ready for library preparation. For library preparations, the samples were amplified with Nextera Index Primers (FC-121-1011) using NEBNext High-Fidelity 2× PCR Master Mix (M0541) and cycled 11 times. After PCR the reactions were cleaned first with the Qiagen MinElute PCR Purification Kit and then with AMPure XP beads (A63881) at a ratio of 1.8:1.0 followed by elution with 15 µl EB.

Adapter sequences and bad quality bases were removed with Cutadapt[57] version 1.16 with options -a CTGTCTCTTATACACATCTCCGAGCCCA CGAGAC -A CTGTCTCTTATACACATCTGACGCTGCCGACGA -q 30 -m 15. Reads were mapped with bowtie[62] 2.4.1 with parameters -I 0 -X 1000 --fr --dovetail --very-sensitive on mm10 or on the II1TDOM-542 mutant genome. Alignments with a mapping quality below 30, discordant pairs, and reads mapping to the mitochondria, were discarded with bamtools version 2.4.1 [https://github.com/pezmaster31/bamtools]. PCR duplicates were removed with [http://broadinstitute.github.io/picard/index.html] version 2.18.2 before the BAM to BED conversion with bedtools[63] version 2.30.0. Coverage and peak calling were computed by macs2[61] version 2.1.1.20160309 with options --format BED --gsize 1870000000 --call-summits --keep-dup all --bdg --nomodel --extsize 200 --shift -100.

**Capture Hi-C.** Samples used in the Capture Hi-C were identified by PCR screening embryos at E12.5 as described above. Collagenase-treated samples were cross-linked with 1% formaldehyde (ThermoFisher 28908) for 10 min at room temperature and stored at −80 °C until further processing. The SureSelectXT RNA probe design used for capturing DNA was done using the SureDesign online tool by Agilent. Probes cover the region mm9 chr2:72240000–76840000 producing 2× coverage, with moderately stringent masking and balanced boosting. Capture and Hi-C were performed as previously reported[64]. Sequenced DNA fragments were processed as previously reported but the mapping was performed on a mutant genome reconstructed from minion and Sanger sequencing (see below). A custom R (www.r-project.org) script based on the SeqinR package[65] was used to construct a FASTA file for the mutant chromosome 2 from the wild-type sequence and the exact position and sequence of breakpoints.

Subtraction of matrices was performed with HiCExplorer[66–68] version 3.6. Heatmaps were plotted using version 3.7 of pyGenomeTracks[66,69].

**II1 TgN cloning and transgenesis.** The II1 enhancer sequence (mm10 chr2:74075305-74075850) was amplified from the fosmid clone WI1-109P4 using primers 001 and 002 (Supplementary Data 1). The 001 primer contains a *XhoI* site and LoxP sequence followed by sequence to the II1 enhancer. The 002 primer contains at its 5′ end a *HindIII* site. This PCR product was gel purified with Qiagen Gel Extraction Kit (28704). The PCR fragment and the pSKlacZ reporter construct were digested with *XhoI* and *HindIII* and ligated together with Promega 2× Rapid Ligation kit (C6711) to produce pSK-II1-LoxP-*LacZ* construct. This vector (15 µg) was cut with *XbaI* and *XhoI* to release the enhancer-reporter construct. The digest was separated on a 0.7% agarose gel for 90 min. The 4190 bp fragment was excised from the gel and purified with Qiagen Gel Extraction Kit (28704) and eluted in 30 µl EB followed by phenol-chloroform extraction and ethanol precipitation and then the pellet was dissolved in 30 µl TE (5 mM Tris pH7.5, 0.5 mM EDTA pH8.0). DNA was injected at 3 ng/µl into pronuclei. Five founder animals were identified carrying the transgene by PCR. Four male founders with the transgene were put into cross with wild-type females and embryos were collected at E12.5 to test for *LacZ* staining. All four male founder lines (*LacZ*/40, 41, 44, and 46) produced distal limb staining but *LacZ*/40 was chosen for amplification of the breeding line due to high transmission of the transgene. For the assembly of the II1 Gfp TgN we used the same construction of the transgene as used for the II1 *LacZ* except that we cloned the GFP coding sequence from pCS2+:Gfp and inserted it using the NcoI and XbaI corresponding restriction sites between II1:HBB and the SV40 poly A signal.

**II1 T-DOM targeted insertion.** The II1 TgN transgenic construct, outlined above, was used for the targeted insertion but homology arms (HA) were attached (Left HA: mm10 chr2:75268556–75269591, Right HA: mm10 chr2:75269617–75270665). The cloning vector was linearized with *KpnI*, separated on a 0.7% agarose gel for 90 min at

90 volts, then the 9.0 kb band was extracted and purified two times with Qiagen Gel Extraction kit (28704). The DNA was quantified by Qubit dsDNA and diluted to 5 ng/µl with IDTE (11-05-01-05). Wild-type fertilized eggs were injected with the construct and the supercoiled pX330 (Addgene #42230) expression vector containing the sgRNA sequence (r4g9: mm10 chr2:75269597–75269616). Animals were genotyped (Supplementary Data 1) to identify founders, and then sequenced with minION (below).

**minION sequencing.** Long-read sequencing was performed on the II1 T-DOM alleles (II1 T-DOM 320 and II1 T-DOM 542) following the nCATS protocol with minor changes[29]. Yolk sacs were isolated from embryos containing the II1 T-DOM transgene and digested with Tail Digestion Buffer (see above) and Proteinase K overnight at 55 °C with no shaking. The following day the samples were incubated at 95 °C for 10 min to inactivate the Proteinase K followed by ethanol precipitation and eluted in 200 µl of 10 mM Tris pH7.5. CRISPR guides (Supplementary Data 1) were designed in CHOPCHOP v3.0[70] and synthesized as Alt-R RNAs by IDT. CRISPR crRNAs were duplexed with tracrRNAs according to the IDT protocol (Alt-R CRISPR-Cas9 System: In vitro cleavage of target DNA with ribonucleo-protein complex, version 2.2). Two master mixes of guide RNAs and Cas9 protein (1081059) were prepared (see Supplementary Fig. 2 for sequence and cutting locations with the locus map), containing either SCS-12 and SCS-13 or SCS-14. The gDNA (9 µg) from the yolk sac was dephosphorylated with NEB Quick CIP (M0510) for 10 min at 37 °C followed by 2 min at 80 °C to inactivate the CIP. The gDNA was split into two equal pools and each pool was then combined with the guide RNP master mixes to cut the gDNA for 30 min at 37 °C followed by 5 min at 72 °C. The samples were then A-Tailed and AMX adapters were ligated (Oxford Nanopore SQK-LSK109). The reactions were size selected with 0.3X Ampure SPRI beads (A63881) followed by two washes with Long Fragment Buffer and then eluted for 30 min in 15 µl EB. The DNA libraries were then prepared according to the Oxford Nanopore protocol for sequencing on a minION (ENR_9084_v109_revP_04Dec2018). The sequencing ran for approximately 24 h and was stopped for processing after all nanopores were depleted.

Bases were called from the fast5 files using Guppy base-caller (Oxford Nanopore Technologies) for CPU version 5.0.16 + b9fcd7b. Reads were mapped on mm10 with minimap2[71] version 2.15 with parameter -ax map-ont. Only primary alignments were kept with samtools view version 1.10[59,60] and reads mapping to II1 (mm10:chr2:74073413-74076528) or to the insertion region (mm10:chr2:75262998–75286118) were further analyzed. Read sequences were compared to the wild-type genome, the expected mutant genome as well as the sequence of the cloning vector using a Perl script as in ref. [72] with the following modification: 20 bp of the MinION reads were tested against the reference for 5 bp-sliding windows and only 20-mers completely identical to unique 20-mers in the reference were kept. The output was then processed in R (www.r-project.org) to display dot plots. The in-depth analysis of reads for allele 320 allowed to propose a configuration that would match all reads containing in total four times the II1-*LacZ* construct.

**Del II1 TFBS TgN cloning and transgenesis.** The enhancer constructs with mutations (Fig. 4c) in the transcription factor binding sites were constructed in silico and synthesized by TWIST Bioscience (San Francisco, CA). The enhancer sequences are available in Supplementary Data 1. The enhancer sequences were synthesized with *BglII* and *AgeI* restriction sites at the 5′ and 3′ ends respectively. The mutant enhancer sequences (Del 2 × 13 and Del 3 × 13) were restriction digested along with pSKlacZ and ligated together with Promega 2X Rapid Ligation kit to produce pSK-II1Del2X13lacZ and pSK-II1Del3X13lacZ. The enhancer-reporter fragments were released from the vector with *BglII* and *XhoI* and purified as above. Pro-nuclear injections were performed by the transgenic platform of the University of Geneva, medical school (CMU). Embryos were collected at approximately E12.5 and stained for *LacZ*.

**Deletions of transcription factor binding sites within T-DOM in vivo.** Guide sequences were selected from the UCSC mm10 genome browser track CRISPR/Cas9 -NGG Targets. The crRNA Alt-R guides were synthesized by IDT. Males homozygous for the II1 T-DOM 542 allele were crossed with super-ovulated wild type females (BL6XCBA-F1) and fertilized eggs were collected. The embryos were electroporated with CRISPR guides (12 µg of each guide) and TrueCut Cas9 v2 protein (Thermo Fisher A36497) with a NEPA21 (NEPA GENE Co. Ltd, Chiba, Japan) and then reimplanted into surrogate females. Embryos were collected at E12.5. Yolk sacs were digested and the II1:*HBB:LacZ* transgene was PCR amplified and Sanger sequenced to identify transgenic embryos containing the mutagenized enhancer element. Embryos that were mosaic for the mutation were not included in this analysis. The embryos were *LacZ* stained (see above) at 37 °C for 16 h, washed in PBS and post-fixed, then stored in 70% ethanol for photographing.

**Generation of Del *Mtx2*-II1-T-DOM and Del II1-T-DOM-*Hnrnpa3* alleles in vivo.** The same methodology was used for this experiment as in the Del II1 T-DOM TFBS experiment above. The sequences for CRISPR guides used in this experiment are listed in Supplementary Data 1. The embryos were genotyped for

the presence of the deletion using primers in Supplementary Data 1. Embryos with ambiguous genotyping results were not used in these results.

**Targeted insertion of Del 3 × 13 and *HBB:LacZ* transgene in T-DOM of ESCs.** The two targeted insertions of control transgenes into T-DOM, genetic editing, and cellular culture were performed as previously reported (Supplementary Fig. 4-2)[73,74]. The guide sequence (r4g9, Supplementary Data 1) was cloned into pX459 vector from Addgene (#62988) and 8ug of the vector was used for mESC transfection. The pX459 vector was co-transfected with 4ug of the vector containing one of the two transgene cassettes. The two transgene cassettes (Del 3 × 13 and HBB:LacZ) contain the same vector backbone with the HBB promoter, LacZ gene, and SV40 polyA signal and homology arms used to target the r4 region of T-DOM (mm10 chr2:75269597-75269616). This is the same insertion site as the II1 T-DOM 542 and 320 alleles. The Del 3 × 13 contains the II1 enhancer element but with the three HOX13 binding sites removed (see Fig. 4c, d). The HBB:*LacZ* transgene is the same but does not contain any portion of the II1 enhancer element so that it is strictly an enhancer sensor in the T-DOM. These constructs were co-transfected into G4 mESCs obtained from the Nagy laboratory[75]. After genotyping to confirm the insertion, the desired mESCs were thawed, seeded on male and female CD1 feeders, and grown for 2 days before the aggregation procedure. ESCs were then aggregated with tetraploid (C57Bl6J × B6D2F1) morula-stage embryos and let developed until blastula prior to transfer into CD1 foster females by the transgenic mouse platform at the University of Geneva Medical School[76].

**Statistics and reproducibility.** ATAC-Seq and HOXA13 and HOXD13 CUT&RUN experiments in Fig. 1a were performed twice on pools of limb tissue from several embryos coming from different litters. The LacZ stained II1 TgN was performed on dozens of individual embryos. In situ hybridizations in Fig. 1c for *Hoxa13* and *Hoxd13* were performed on three wild-type embryos for each probe. The II1 Gfp embryos in Supplementary Fig. 1-1b are a stable breeding line so dozens of embryos were evaluated. The in situs in Supplementary Fig. 1-2b were performed on three different embryos of each genotype and for each probe. In Fig. 2b, c, LacZ staining was performed on dozens of embryos of each genotype and across developmental stages. In Fig. 3a, ATAC-Seq, HOXA13 and HOXD13 CUT&RUN experiments were performed twice, on separate pools of limb tissue from several embryos coming from different litters. One replicate is plotted as a solid color histogram, and the second replicate is plotted as a dark gray line in the same panel. In Fig. 4b, dozens of control embryos were stained for LacZ (II1 T-DOM control). Embryos injected with g2 + g3 with altered staining and confirmed genotypes, $n = 13$, $n = 16$ for g2 + g6 embryos, and $n = 6$ embryos containing the large deletion Del C-T g2 + g6. See images in Supplementary Fig. 4-1. The number of embryos used for Fig. 4d is indicated in the table in that panel. Also, see Supplementary Fig. 4-2. All of the embryos used for Fig. 5b, c are shown in Supplementary Fig. 5, $n = 6$, $n = 5$, $n = 5$, and $n = 6$ for each confirmed genotype. Capture Hi-C experiments in Fig. 6 were performed on a single pool of tissue from several embryos of the same genotype.

**Reporting summary**. Further information on research design is available in the Nature Research Reporting Summary linked to this article.

# Data availability
The data that support this study are available from the corresponding author upon reasonable request. All sequencing data generated in this study have been deposited in the Gene Expression (GEO) database under accession code GSE194114. Datasets from previous publications include: GSM2061123, GSM2061125, GSM2151013, GSM2151014, GSM2713703, GSM2713704, GSM2713707, GSM2713708. Source data are provided with this paper.

# Code availability
All scripts necessary to reproduce figures from raw data are available at https://github.com/lldelisle/scriptsForBoltEtAl2022[77].

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

## Acknowledgements

We thank Jozsef Zakany for his help in an initial phase of this work and other colleagues from the Duboule laboratories for discussions. This work was supported in part by using the resources and services of the Gene Expression Research Core Facility (GECF) at the School of Life Sciences of EPFL and the transgenic platform at the medical school, University of Geneva. C.C.B was supported by the Eunice Kennedy Shriver National Institute of Child Health & Human Development of the National Institutes of Health, under Award Number F32HD093555. This work was supported by funds from the Ecole Polytechnique Fédérale (EPFL, Lausanne), the University of Geneva, the Swiss National Research Fund (No. 310030B_138662 and 310030_196868 to D.D. and No. PP00P3_176802 to G.A.), and the European Research Council grant RegulHox (No 588029) (to D.D.). Funding bodies had no role in the design of the study and collection, analysis and interpretation of data, and in writing the manuscript.

## Author contributions

C.C.B.: designed and conducted experiments, analyzed datasets, formalized results, and wrote the paper. L.L.-D.: analyzed and evaluated the statistical significance of datasets. Wrote the paper. B.M.: produced and maintained all the mouse lines. A.H.: performed the Del 2 × 13 and 3 × 13 experiment A.R.: performed the Del 3 × 13 T-DOM experiment G.A.: designed and performed the HBB:LacZ T-DOM experiment D.D.: designed experiments, transported mice, dissected some limb buds, and wrote the paper.

## Competing interests

The authors declare no competing interests.
