## [Peer Review File · Nature Communications]

REVIEWER COMMENTS

Reviewer #1 (Remarks to the Author):

The manuscript by Bolt et al uses current genome editing techniques in mouse with lovely cell biology to compare the action of an enhancer (II1) driving Hox gene expression in the distal limb bud in its normal position and in a different context (a regulatory domain normally driving proximal gene expression). The enhancer is shown to work well when randomly integrated in the genome but must show some redundancy as its deletion has no effect on Hoxd transcription. However, in its targeted position II1 is unable to overcome local signals and becomes expressed proximally and silenced distally, despite continuing to be able to bind transacting HOX13 TFs. When Hox binding sites within II1 are deleted, the expression driven by it is also likely to show a proximal pattern. Removal of large sections of flanking DNA alters the genomic context and restores distal expression. Finally, the contacts this integrated transgene makes with the gene cluster is investigated by HiC and is shown to change in proximal and distal mesenchyme, but never recreate the contacts it would have endogenously.

I have only a few minor comments-

As II1 overlaps with a peak identified by Seth at E11.5 and that lines carrying II1 TgN were made, could a timecourse of lacZ staining from these mice be included in Figure 2 to confirm (or otherwise) that the early expression seen in the II1 T-dom lines arises from enhancer sensing rather intrinsic enhancer activity?

Given how nicely the in vivo crispr deletion of TFBS in the T-Dom 542 allele experiment worked and how many independent embryos were generated, I'm not sure what additional information was generated by subsequently integrating the mutated transgene and examining one line. It seems this experiment could be omitted.

(Not of relevance to the paper, but for my interest- why were some transgenics made by microinjection, some by electroporation and others in ES cells?)

The arrows in Figure 5c are missing.

Reviewer #2 (Remarks to the Author):

In this manuscript, authors studied the context-dependency of enhancer by using the unique transgenic setup in the HoxD locus.

The HoxD gene cluster is under the regulation of two different TAD, T-DOM and C-DOM. The T-DOM has enhancers working in the early limb bud and activate the genes in the proximal limb. During the limb bud outgrowth, C-DOM starts controlling the 5' HoxD genes in the distal limb. From the ATAC-seq and Hox proteins ChIP-seq data, they chose the strong distal limb enhancer (named II1) located in the C-DOM and inserted this along with the LacZ reporter gene into the T-DOM. This idea is interesting and the HoxD locus is an ideal locus to test the enhancer activity in the ectopic TAD context.

However, this II1-LacZ showed strong LacZ activity in the proximal limb and its activity in the distal limb was very weak, even the II1 has strong activity in the natural context in C-DOM and random location from the transgenic assay. The recombined II1 enhancer was not

active due to the repressive condition of T-DOM in the distal limb, and the LacZ reporter gene became an enhancer sensor in the proximal limb.

The authors focused on the weak activity in the distal limb and confirmed Hoxa13/d13 proteins bound to the II1 in the distal forelimb. Then they deleted of binding sites of these transcriptional factors. The weak activity in the distal limb was completely disappeared in Del-TFBS. Thus, this weak activity depends on the Hox proteins. They also made large deletion (Del Mtx2-II1) in the T-DOM and confirmed the II1 enhancer activity recovered in the distal limb.

Finally, they checked the chromatin structure by capture Hi-C. Surprisingly, the recombined II1 T-DOM transgene showed a stronger interaction with HoxD genes. However, it cannot overcome the barrier of T-DOM TAD and never contacted the Hoxd12 to Hoxd13 region.

The strengths of this paper are the unique experiment setting and admirable mouse genetics works. The HoxD locus is ideal for testing the enhancer activity in the ectopic TAD context. Targeting deletion of Hox protein binding sites in the transgene locus will take time and effort. A series of detailed analysis of mutant mice using the multiple genomics method thoroughly explains the situation of the II1 T-DOM transgene. This transgene showed no stable ATAC-seq peak, however, interacting with the HoxD cluster in the proximal limb is interesting. Multiple copies of the transgene (line 320) showing the strong activity in the distal limb indicates that a powerful enhancer can overcome the TAD structure.

However, their main finding, TAD mediated silencing of enhancer in a specific tissue, was already described in another locus, such as the Pitx1 and its enhancer Pen (<https://doi.org/10.1038/s41588-018-0221-x>). Their conclusion for the context-dependency of enhancer is ambiguous and it's hard to generalize. This manuscript would be of interest to HoxD researchers, but I am not sure for other researchers.

Major concerns.

1: Authors concluded that this transgene is a new proximal enhancer based on capture Hi-C data, but I am not sure if this transgene activates the 3' HoxD genes in the proximal limb. Due to the resolution and DpnII site problem, they cannot discriminate if the II1 or HBB promoter mediates this interaction. I think qPCR data for the HoxD genes in the proximal limb of the recombined II1 T-DOM transgenic mice will be needed.

2: If this transgene acts as a new proximal enhancer, what protein mediates the interaction between the II1 transgene and HoxD genes in the proximal limb? Have you checked the Hox protein binding at the proximal limb?

3: I am rather interested in the multicopy transgenic mice (line 320) as this mouse overcome the T-DOM silencing effect in the distal limb. Any mechanistic insight (histone modifications, chromatin structure) to overcome the TAD barrier would be great as a follow-up study in the near future.

Minor comments:

Missing the scheme figure (Fig S4 1-C)

Reviewer #3 (Remarks to the Author):

The study by Bolt and colleagues aims to gain insight into how the intra-TAD environment of genes co-regulated by multiple enhancers may impact the activity of a particular

enhancer. They address this important issue by transferring a strong enhancer of the distal expression of HoxD genes in mouse limbs from the parental C-DOM TAD into the T-DOM TAD, which controls the proximal expression of HoxD genes. The transgene insertion into the T-DOM TAD consists of the II1 enhancer linked to a LacZ reporter construct with a minimal promoter. This insertion results in the II1 enhancer losing its distal activity almost completely and gaining strong proximal LacZ activity. Combining ChIP analysis of HOX13 proteins with in cis-deletion analysis of the T-DOM TAD, chromatin structure analysis and other molecular analysis, the authors show that the normal distal II1 enhancer activity is suppressed by the change to the T-DOM chromatin context, which exerts a dominant effect imposing proximal activity. The analysis is comprehensive and of high technical quality. In particular, the capture-HiC analysis shows that the II1 enhancer in the T-DOM TAD becomes part of the contacts that the other proximal enhancers establish with the HoxD gene cluster.

This reviewer has one major issue with the conclusions. It appears that the authors interpret the data such that the II1 enhancer inserted into the T-DOM TAD loses its strong distal activity and gains proximal activity. It is this latter point that in this reviewer's opinion is not established for the following reasons. As II1 enhancer activity in the T-DOM domain is assayed indirectly by LacZ, it is possible II1 enhancer retains only low distal activity, but the strong proximal LacZ reporter expression is due to cis-regulation by the "endogenous" T-DOM enhancers. This is also a feasible interpretation for the results of the deletion analysis shown in Figure 5. Therefore, the II1 enhancer may not gain any proximal activity even in a chromatin context that brings it in close contact with proximally expressed Hoxd genes.

The authors should address this experimentally as based on the current analysis one cannot make a case for context dependent positive enhancer functions in the sense implied by both title and abstract. The simplest interpretation is that the II1 enhancer loses most of its activity upon translocation and the proximal enhancers in T-DOM impose proximal LacZ reporter expression independent of II1, i.e. there is only loss but no gain of II1 enhancer functions upon its insertion into the T-DOM TAD.

Additional points:

Fig. S4-1- S4-3: This supplementary Figure is in three parts and confusing because panel labels in text and figures do not match or are missing. In particular a scheme (Fig. S4-1C) very important for the understanding of what was done (text lines: 329-241) is missing. This reviewer recommends careful revisions of these figures and separating them into three separate supplementary figures.

In the text, the authors refer to 3' and 5' Hox genes at some stage and also this is often used in publications. Therefore it would help understanding if in all schemes of the Hox TADs the 3'-5' direction would be indicated.

**REVIEWER COMMENTS**

**Reviewer #1 (Remarks to the Author):**

*The manuscript by Bolt et al uses current genome editing techniques in mouse with lovely cell biology*
*to compare the action of an enhancer (III) driving Hox gene expression in the distal limb bud in its*
*normal position and in a different context (a regulatory domain normally driving proximal gene*
*expression). The enhancer is shown to work well when randomly integrated in the genome but must*
*show some redundancy as its deletion has no effect on Hoxd transcription. However, in its targeted*
*position III is unable to overcome local signals and becomes expressed proximally and silenced*
*distally, despite continuing to be able to bind transacting HOX13 TFs. When Hox binding sites within*
*III are deleted, the expression driven by it is also likely to show a proximal pattern. Removal of large*
*sections of flanking DNA alters the genomic context and restores distal expression. Finally, the contacts*
*this integrated transgene makes with the gene cluster is investigated by HiC and is shown to change in*
*proximal and distal mesenchyme, but never recreate the contacts it would have endogenously.*

*I have only a few minor comments-*

*As III overlaps with a peak identified by Seth at E11.5 and that lines carrying III TgN were made,*
*could a timecourse of LacZ staining from these mice be included in Figure 2 to confirm (or otherwise)*
*that the early expression seen in the III T-dom lines arises from enhancer sensing rather intrinsic*
*enhancer activity?*

Yes, we agree with the reviewer that this is an important detail to understand the normal
behavior of the III enhancer and we appreciate the suggestion. We had initially (for different
purposes) established two sets of transgenic lines with the same piece of the III enhancer DNA
but including either the *LacZ* (stronger staining, more precise) or the *GFP* (dynamic
observation, more labile, less protein accumulation effect) genes as reporters. To answer the
referee's comment, we have now introduced a new supplementary figure related to Figure 1
(Figure S1-1) by using the *GFP* line. In this supplementary Figure S1-1B (line 182) we now
show a time-course of the *GFP* transgene activity, which we believe clarifies this issue. Indeed,
the transgene exactly corresponds to the *Hoxd13* pattern in limbs, with no detectable expression
in posterior cells where T-DOM enhancers are active, even in E10 limb buds. In addition, it
provides yet another illustration of 1) the strength of this enhancer sequence (*GFP* staining is
rarely that strong at this tissue scale) and 2) the amazing specificity of this sequence for distal
limb development.

*Given how nicely the in vivo crispr deletion of TFBS in the T-Dom 542 allele experiment worked and*
*how many independent embryos were generated, I'm not sure what additional information was*
*generated by subsequently integrating the mutated transgene and examining one line. It seems this*
*experiment could be omitted.*

This experiment provides an additional control for two different experiments in order to unify
their results. First, while the result of the randomly integrated Del 3x13 transgene embryos
(Figure 4D) is clear and indicates that it cannot activate transcription of the *LacZ*, this result

still formally does not demonstrate that it may not do it when inserted into T-DOM, since we
precisely like to look -in this paper- at the effect of the TAD environment upon enhancer
sequences. We agree that the result is not unexpected, yet we think it is a necessary control.
Second, it provides an orthogonal approach to the conclusion of the CRISPR-cut III T-DOM
542 alleles (Figures 4A, B and S4-1), because the sequences of the CRISPR-cut III elements
do not exactly (i.e., to the base) match the sequence of the 3x13 construction. We wanted to
confirm that the CRISPR-cut version does not create anything like a synthetic binding site or
whatever, potentially recognized by some repressive factor that could produce an ambiguous
result. There again, this would be a very unlikely situation, yet something that could be easily
addressed by using this control and hence we think that this simple result should remain in the
manuscript.

*(Not of relevance to the paper, but for my interest- why were some transgenics made by microinjection,*
*some by electroporation and others in ES cells?)*

These different approaches each have their own purposes and limitations. For example, to
generate random transgenesis models (e.g. III:*LacZ* TgN and III:*Gfp* TgN, Figure 1C and S1-
1B) or the targeted insertion of the transgene line III T-DOM 542, the transgene material is
dsDNA. Electroporation does not (or very poorly -our experience-) carry naked dsDNA into
zygotes so pronuclear injection was necessary. After the 542 allele was established, we wanted
to modify the allele using the CRISPR system. The CRISPR components (Cas9 and gRNA)
can be carried into zygotes very efficiently by electroporation in the form of RNA and protein.
This system is highly efficient in our hands (and so much simpler to implement) making it
possible to do large scale genetic manipulations (e.g., Figure S4-1 and S5). We generated the
control embryos (Figure S4-3) in ESCs followed by tetraploid aggregation because this
technique allows for efficient and reproducible testing of these transgene constructs without
the need to generate stable breeding lines (saving time). It would be much more difficult to
generate these controls by pronuclear injection, and for the purposes of this experiment we did
not need to establish stable lines. In fact, the overall strategy is to try to systematically select
the best appropriate approach to maximize efficiency, while minimizing the use of animals.

*The arrows in Figure 5c are missing.*

Thank you for noticing. We have now corrected this omission.

**Reviewer #2 (Remarks to the Author):**

*In this manuscript, authors studied the context-dependency of enhancer by using the unique transgenic*
*setup in the HoxD locus. The HoxD gene cluster is under the regulation of two different TAD, T-DOM*
*and C-DOM. The T-DOM has enhancers working in the early limb bud and activate the genes in the*
*proximal limb. During the limb bud outgrowth, C-DOM starts controlling the 5' HoxD genes in the*
*distal limb. From the ATAC-seq and Hox proteins ChIP-seq data, they chose the strong distal limb*
*enhancer (named III) located in the C-DOM and inserted this along with the LacZ reporter gene into*
*the T-DOM. This idea is interesting and the HoxD locus is an ideal locus to test the enhancer activity*
*in the ectopic TAD context.*

*However, this III-LacZ showed strong LacZ activity in the proximal limb and its activity in the distal*
*limb was very weak, even the III has strong activity in the natural context in C-DOM and random*
*location from the transgenic assay. The recombined III enhancer was not active due to the repressive*
*condition of T-DOM in the distal limb, and the LacZ reporter gene became an enhancer sensor in the*
*proximal limb.*

*The authors focused on the weak activity in the distal limb and confirmed Hoxa13/d13 proteins bound*
*to the III in the distal forelimb. Then they deleted of binding sites of these transcriptional factors. The*
*weak activity in the distal limb was completely disappeared in Del-TFBS. Thus, this weak activity*
*depends on the Hox proteins. They also made large deletion (Del Mtx2-III) in the T-DOM and*
*confirmed the III enhancer activity recovered in the distal limb.*

*Finally, they checked the chromatin structure by capture Hi-C. Surprisingly, the recombined III T-*
*DOM transgene showed a stronger interaction with HoxD genes. However, it cannot overcome the*
*barrier of T-DOM TAD and never contacted the Hoxd12 to Hoxd13 region.*

*The strengths of this paper are the unique experiment setting and admirable mouse genetics works. The*
*HoxD locus is ideal for testing the enhancer activity in the ectopic TAD context. Targeting deletion of*
*Hox protein binding sites in the transgene locus will take time and effort. A series of detailed analysis*
*of mutant mice using the multiple genomics method thoroughly explains the situation of the III T-DOM*
*transgene. This transgene showed no stable ATAC-seq peak, however, interacting with the HoxD*
*cluster in the proximal limb is interesting. Multiple copies of the transgene (line 320) showing the strong*
*activity in the distal limb indicates that a powerful enhancer can overcome the TAD structure.*

*However, their main finding, TAD mediated silencing of enhancer in a specific tissue, was already*
*described in another locus, such as the Pitx1 and its enhancer Pen ([https://doi.org/10.1038/s41588-](https://doi.org/10.1038/s41588-018-0221-x)*
*018-0221-x). Their conclusion for the context-dependency of enhancer is ambiguous and it's hard to*
*generalize. This manuscript would be of interest to HoxD researchers, but I am not sure for other*
*researchers.*

*We thank the reviewer for these nice comments, yet we do have a different appreciation of the*
*last point regarding the similarities with the work published by Kragestein et al. (Nature*
*Genetics, 2018), a paper that, by the way, was last-authored by an author of this submitted*
*manuscript (and hence there is little chance that we misunderstood an important issue).*

*First, the reviewer summarizes our main finding as "TAD mediated silencing of enhancer in a*
*specific tissue" and states that this finding has already been described in the Kragestein et al.*
*article. In a superficial sense, there is indeed some similarity between the take home message*
*of our manuscript and the Pitx1 study. The Pitx1 locus and our III T-DOM transgene both*
*depend on the complete global TAD structure in order to implement the repression of their*
*respective genes. However, we think that this is the limit of the similarities between these two*
*studies, for they provide clearly distinct (and useful) information to the field of gene regulation.*

*To quote the Pitx1 study: "The required tissue specificity of the enhancer is facilitated by*
*modifying its position in the 3D chromatin space." The authors conclude this because at the*

*Pitx1* locus, the change between transcriptionally productive (hindlimbs) and unproductive
(forelimb) states depends on a reconfiguration of the locus - global restructuring of the
chromatin fiber in three-dimensional space. In both forelimbs and hindlimbs the Pen enhancer
is in an active state (see Figure 3C in *Kragesteen et al.*), so there is no modulation of the Pen
enhancer *per se* in these two tissues. However, in the forelimbs, the Pen enhancer is not able
to productively contact the *Pitx1* promoter because the entire locus is configured to sequester
the Pen enhancer into forelimb-specific 3D configuration (see Figure 4). This observation is
supported by the experiment in Figure 1 using a *LacZ* sensor integrated at two different
locations in the locus; when integrated near the *Pitx1* promoter the *LacZ* reports only in the
hindlimbs, but when integrated near to the Pen enhancer it reports in forelimbs and hindlimbs.

What we observe in our study is **fundamentally** different. First, in the proximal and distal limb
segments, the T-DOM three-dimensional organization largely remains the same between these
tissues. The T-DOM 3D interaction profile only changes very slightly between the proximal
and distal limb environments indicating that there are no global and very few weak specific
reconfigurations of the T-DOM (Figure 6, and see Rodriguez-Carballo 2015, Figure 1D).
Indeed, we see that the III T-DOM transgene follows the existing tropism of the TAD, making
contacts with the *Hoxd* gene cluster in the same general way that the remainder of the T-DOM
does in either tissue type. Second, while the III T-DOM enhancer is not active in proximal
limb cells, it does show some hallmarks of an active enhancer in distal limb cells (Figure 3).
In this case, unlike in *Kragesteen et al.*, the III T-DOM enhancer is inactivated precisely in the
distal cells where it (as a distal enhancer) should normally be active (because it is now in T-
DOM). In the absence of 3D changes that sequester the III enhancer element away from the
*LacZ* transgene (a distance of 400-500bp), we think that this may point towards the T-DOM
somehow excluding some factor that is needed to couple HOX13 transcription factors activities
to the transgene promoter, though other explanations are possible.

In *Kragesteen et al.* the enhancer remains in place with the same activity but its communication
to its cognate gene is blocked in a specific tissue. In *Bolt et al.*, the enhancer is transferred into
a TAD with a different specificity and, as a consequence, is prevented to work where it should.
We are not aware of any published experiments of this kind and hence we continue to think
that this work will interest the entire community of colleagues working in gene
regulation/development/chromatin.

*Major concerns.*

*I: Authors concluded that this transgene is a new proximal enhancer based on capture Hi-C data, but*
*I am not sure if this transgene activates the 3' HoxD genes in the proximal limb. Due to the resolution*
*and DpnII site problem, they cannot discriminate if the III or HBB promoter mediates this interaction.*
*I think qPCR data for the HoxD genes in the proximal limb of the recombined III T-DOM transgenic*
*mice will be needed.*

We would like to apologize to the reviewers for an unfortunate lack of clarity from our part in
the 'Conclusion' section of the manuscript only, which likely triggered this comment. This
mistake may give the feeling that our conclusion is exactly the opposite of what it actually is.

In the initial manuscript, the relevant text reads (line 577): “*When this distal enhancer was*
*introduced into this ‘proximal TAD’, it behaved in all respects like its new neighbor proximal*
*enhancers, thus illustrating the potential of chromatin domains, in some cases, to impose*
*another level of coordinated regulation on top of enhancer sequence specificities.*” Based on
this text, we understand why the reviewer thinks our conclusion is that the III T-DOM enhancer
acquires proximal limb enhancer activity, especially within the context of the ‘Conclusion’
paragraph as written, which conflates the 3D contacts with causal enhancer activity. We have
of course modified the ‘Conclusion’ section to correct this source of confusion and we have
made an effort to improve the clarity of the text throughout the manuscript (lines 273, 394,
476, 608, 619). Of note, however, in all other places in the original manuscript, we were careful
to clearly indicate that the proximal expression of *LacZ* was due to enhancer trapping, because
of the proximity of ‘real’ proximal enhancers and that the III T-DOM enhancer element is
decommissioned in the distal limb by the T-DOM. Indeed, we state at several places in the text
that the proximal limb *LacZ* staining is the result of the *LacZ* sensing the native T-DOM
proximal limb enhancer activity (see lines 224, 252, 256, 278, 340, 360, 371, 388). The wording
of the ‘Conclusion’ section was not as precise, we agree.

With regard to the capture Hi-C and gene expression, we whole-heartedly agree with the
reviewer’s statements. We have added additional text to clarify that the III T-DOM enhancer
is probably not responsible for any of the transcriptional output of the *Hoxd* genes in the
proximal or distal limb (line 478, 620). The contacts made between the III T-DOM transgene
and the gene cluster result from something akin to a hostage situation: the T-DOM normally
makes contacts with the cluster; globally the T-DOM contacts with the cluster do not change
in the presence of the III T-DOM transgene; since the transgene is embedded in this
environment, it is carried by the T-DOM into contact with the gene cluster. However, these
contacts are most likely not productive for two reasons: the III enhancer in the T-DOM is not
active in the proximal limb (Figure 1C, S1, 3), and it is repressed by the T-DOM in the distal
limb.

We hope that this clarification and the changes to the text are sufficient to address the
reviewer’s concerns. We think that additional experiments will not provide better clarity than
improvements to the text.

*2: If this transgene acts as a new proximal enhancer, what protein mediates the interaction between*
*the III transgene and HoxD genes in the proximal limb? Have you checked the Hox protein binding at*
*the proximal limb?*

Based on the experiments in this study, we do not observe the III T-DOM enhancer acting as
a new proximal limb enhancer (see above). We did not evaluate binding of any transcription
factors in the proximal limb because we do not have any evidence that the III enhancer is active
in there, and, rather, we do have evidence that the III enhancer is not active in the proximal
limb (Figure 3, S4-3). The *LacZ* staining observed in the proximal limb is the result of *LacZ*
sensor responding to the proximal limb enhancers that natively reside within the T-DOM (see
above).

*3: I am rather interested in the multicopy transgenic mice (line 320) as this mouse overcome the T-*
*DOM silencing effect in the distal limb. Any mechanistic insight (histone modifications, chromatin*
*structure) to overcome the TAD barrier would be great as a follow-up study in the near future.*

We agree with the reviewer that this is a very interesting observation and we would love to
follow-up with additional experiments. However, as we briefly mentioned in the manuscript
(lines 244, 543), these experiments are nearly impossible to analyze. The reason is because the
sequence of the transgene is present in the genome as several copies. Any sequencing reads
produced by chromatin-based assays wouldn't be uniquely assignable to any one of these
copies. That makes it very difficult to identify any modifications would be distributed over the
transgenic DNA and hence to propose a hypothesis regarding how these multiple copies may
affect chromatin structure etc. Similarly, CRISPR-based mutagenesis experiments would rely
on guide sequences that would be present in each copy of the transgene. Any Cas9 that finds a
target in one transgene copy will find the same target sequence in the other transgenes. There
would be no way to parse which mutation is responsible for any outcome. For these reasons
we will not be performing follow-up experiments on the 320 allele. A conclusion from looking
at this line, however, is that transgenic integration should be very carefully analyzed for their
copy number as this factor may be determinant for the behavior of a reporter gene.

*Minor comments:*

*Missing the scheme figure (Fig S4 I-C)*

We apologize. The version of Figure S4-1 included in the manuscript did not include this
scheme, which makes the reading of the paper easier. The correct version of this figure is
included in the revised manuscript and we have also included it in this document (see below)
for a direct evaluation by the reviewers.

**Reviewer #3 (Remarks to the Author):**

*The study by Bolt and colleagues aims to gain insight into how the intra-TAD environment of genes co-*
*regulated by multiple enhancers may impact the activity of a particular enhancer. They address this*
*important issue by transferring a strong enhancer of the distal expression of HoxD genes in mouse*
*limbs from the parental C-DOM TAD into the T-DOM TAD, which controls the proximal expression of*
*HoxD genes. The transgene insertion into the T-DOM TAD consists of the III enhancer linked to a*
*LacZ reporter construct with a minimal promoter. This insertion results in the III enhancer losing its*
*distal activity almost completely and gaining strong proximal LacZ activity. Combining ChIP analysis*
*of HOX13 proteins with in cis-deletion analysis of the T-DOM TAD, chromatin structure analysis and*
*other molecular analysis, the authors show that the normal distal III enhancer activity is suppressed*
*by the change to the T-DOM chromatin context, which exerts a dominant effect imposing proximal*
*activity. The analysis is comprehensive and of high technical quality. In particular, the capture-HiC*
*analysis shows that the III enhancer in the T-DOM TAD becomes part of the contacts that the other*
*proximal enhancers establish with the HoxD gene cluster.*

*This reviewer has one major issue with the conclusions. It appears that the authors interpret the data*
*such that the III enhancer inserted into the T-DOM TAD loses its strong distal activity and gains*
*proximal activity. It is this later point that in this reviewer's opinion is not established for the following*
*reasons. As III enhancer activity in the T-DOM domain is assayed indirectly by LacZ, it is possible III*
*enhancer retains only low distal activity, but the strong proximal LacZ reporter expression is due to*
*cis-regulation by the "endogenous" T-DOM enhancers. This is also a feasible interpretation for the*
*results of the deletion analysis shown in Figure 5. Therefore, the III enhancer may not gain any*
*proximal activity even in a chromatin context that brings it in close contact with proximally expressed*
*Hoxd genes.*

*The authors should address this experimentally as based on the current analysis one cannot make a*
*case for context dependent positive enhancer functions in the sense implied by both title and abstract.*
*The simplest interpretation is that the III enhancer loses most of its activity upon translocation and the*
*proximal enhancers in T-DOM impose proximal LacZ reporter expression independent of III, i.e. there*
*is only loss but no gain of III enhancer functions upon its insertion into the T-DOM TAD.*

See our answer to referee II. We completely agree with the reviewer's conclusion. We think
that the Conclusion section was not clear and it gave the impression that our conclusion was
what this expert mentions. As explained above, there were eight other places in the manuscript
where the logical and commonly accepted conclusion (enhancer trapping) was correctly and
clearly proposed. The lack of precision of this single statement in the 'Conclusion' section was
problematic and escaped our last reading. Sorry for this.

*Additional points:*

*Fig. S4-1- S4-3: This supplementary Figure is in three parts and confusing because panel labels in text*
*and figures do not match or are missing. In particular a scheme (Fig. S4-1C) very important for the*
*understanding of what was done (text lines: 329-241) is missing. This reviewer recommends careful*
*revisions of these figures and separating them into three separate supplementary figures.*

Sorry for this too. The version of Figure S4-1 included in the manuscript was lacking this
scheme, which is shown only to facilitate the understanding of this experimental configuration,
whereas it contains no result in itself. The correct version of this figure is now included in the
revised manuscript, and we have also included it below for easy evaluation by the reviewer.

*In the text, the authors refer to 3' and 5' Hox genes at some stage and also this is often used in*
*publications. Therefore, it would help understanding if in all schemes of the Hox TADs the 3'-5'*
*direction would be indicated.*

Thanks for the suggestion. we have added a modification to the schemes to indicate the
direction of transcription and added text to the figure legends to clarify this.

A

Del TFBS

B

Del C - T

C

REVIEWERS' COMMENTS

Reviewer #1 (Remarks to the Author):

In response to my comments, the authors have added a time course of very pretty GFP transgenics to show the activity of II1 enhancer.

I am happy with the changes made to the manuscript and would recommend publication.

Reviewer #2 (Remarks to the Author):

As the authors changed their conclusion and my concerns were solved.

I thus support the publication of this manuscript in Nature Communications.

The line numbers mentioned in the rebuttal letter don't fit both new and old manuscript line numbers.

For example, #608, 619 is in the methods section.

This makes it hard to compare side by side between old and new manuscripts, though I read the whole manuscript and it looks fine to me.

Reviewer #3 (Remarks to the Author):

The authors have addressed my concerns appropriately such that publication of the manuscript can now be recommended.